# Home bias and employee social responsibility: Identification vs. benefit exchange

**Tingting Zhang[1], Wenjing Sun[2], Xing Rong**  **[1]\*, Jingyi Mu[1]**

**1** School of Finance, Southwestern University of Finance and Economics, Chengdu, Sichuan, China,
**2** School of Accounting, Southwestern University of Finance and Economics, Chengdu, Sichuan, China

\* rx865@126.com

**Data Availability Statement:** All relevant data are within the article and its Supporting Information files.

**Funding:** YES This work was partially supported by the National Science Foundation for Young

## Abstract

Employees, as the most valuable assets and critical sources of competitive advantage in enterprises, are among the important stakeholders in enterprises. Employee social responsibility (ESR) has been a continually important research interest in the field of enterprise social responsibility. However, in the literature, few studies explore how personal features affect employee social responsibility. Thus, sampling China's listed companies from 2006 to 2019, we investigate how the home bias of senior executives influences enterprises' employee social responsibility. We identify home bias based on whether a chairperson's or CEO's hometown matches the firm's registration place. Three main results are obtained. First, the home bias of both CEOs and chairpersons can improve the corporate fulfillment of employee social responsibility. Second, further cost-benefit analysis shows that this result is due to not only identification but also benefit exchange. Although senior executives' home bias significantly decreases employee turnover rate, enterprises absorb more employment, which significantly increases their redundant personnel costs. Therefore, firms should balance the potential benefits and costs incurred by home bias via trade-off. Third, in firms facing less market competition, firms with more governmental subsidies or state-owned firms, senior executives' home bias has a more significant promoting effect on the fulfillment of ESR, supporting the view of benefit exchange. Accordingly, by extending theories on the effects of senior executives' home bias and enriching the ESR literature, this paper has important practical value, our findings can guide and promote firms to perform ESR while actively complying with a national policy for stabilizing employment and ensuring people's well-being.

## 1. Introduction

Employees are the main stakeholders of enterprise social responsibility. The protection of employee rights and interests is the most direct and important part of enterprise social responsibility [1] and has gained extensive attention from international organizations, such as the International Labor Organization and the International Standardization Organization, which aim to standardize enterprise social responsibility [2]. Employee social responsibility is mainly reflected in the care and concerns of firms for their employees. For instance, firms can perform social responsibility by providing employees with competitive salaries while ensuring their

Scientists of China [grant No. 71902163], the Humanities and Social Sciences Youth Foundation of Ministry of Education of China [grant No. 19YJC630218] and Fundamental Research Funds for the Central Universities [grant No. JBK2202015].

**Competing interests:** The authors have declared that no competing interests exist.

welfare and safe working conditions [3] or by offering employees sufficient training and humane treatment [4]. By treating employees well and securing their rights and interests, firms can not only promote more employee organizational citizenship behavior [2] and innovation [5] but also achieve higher total factor productivity(TFP) [6], improve future performance [6] and obtain higher growth potential [7].

However, compared to donations, product services and other social responsibilities that the media and public are familiar with, employee social responsibility is more internal and tends to be ignored by firms. The basic performance of employee social responsibility in many firms is rather unsatisfactory. International Labor Organization (ILO) data show that since 2010, there have been at least 44000 work stoppages worldwide; however, more seriously, these data may not cover all economic activities and all geographic areas. The number of disputes in Chinese companies has steadily risen, while in many countries and regions of the world, labor-related conflicts and mobilizations are becoming ever more intense and frequent [8]. In China, according to statistics of the Ministry of Human Resources and Social Security, by the end of 2021, there had been approximately 2.631 million labor disputes, and this number had increased by 140.27% on a year-on-year basis. In these disputes, most problems concern poor working conditions, employee health and safety, limited development potential and unsatisfactory welfare. It is an unarguable fact that disputes arising from labor-management conflicts remain at a high level. At the same time, data also reveal that there are still a large number of companies globally that seriously lack any awareness of consciously fulfilling employee social responsibility. The current situation thus not only restricts the sustainable development of enterprises but also poses a great challenge to securing people's livelihood and maintaining social harmony and stability. Therefore, in-depth research on the fulfillment of employee social responsibility has become an important topic that currently has both theoretical significance and practical value.

In recent years, academics have attempted to distinguish employee social responsibility and general social responsibility for specialized research. The literature has confirmed that market competition [9, 10], regional Confucian cultural intensity [11], corporate hypocrisy [12], the establishment of labor-management committees [13], population mobility [14], and family business restructuring [3] are important factors influencing the fulfillment of employee social responsibility. Unfortunately, most of these studies have been based on external factors, while explorations at the individual level have been less common. Higher-order theory emphasizes that a team of senior executives plays a central role in an organization and that the personal features of these team members have a significant impact on corporate decisions [15]. Therefore, an important question arises: when senior executives are making decisions on employee social responsibility, does their personal bias affect responsibility fulfillment?

Home bias represents an individual's inherent favoritism toward his or her hometown and has been widely discussed worldwide. Governmental officers tend to distribute more political resources to their hometown [16]. Fund managers generally hold stocks of firms from their hometown in a greater proportion than average [17]. Senior executives are inclined to establish more subsidiary firms [18] and make more acquisitions and investments [19–21] in their hometown. However, the literature remains mostly based on the identification economics proposed by Akerlof and Franton [22, 23]. Identification theory emphasizes individual morality and group responsibility but ignores the social exchanges and interest games driven by profit-seeking instincts among individuals and groups. Because the fulfillment of employee social responsibility cannot benefit firms instantaneously, it is often ignored by the media as well as the public, it is valuable to determine the possible motivations for such performance and to explore how firms engage in trade-off during their interest games with local government and other stakeholders.

Based on the above analysis, we investigate the influences and inner motivations of senior executives regarding the corporate fulfillment of employee social responsibility from the perspective of home bias. We attempt to resolve the following questions: First, how does the home bias of senior executives influence the corporate fulfillment of employee social responsibility? Second, what are the inner motivations driving senior executives' influence on the corporate fulfillment of employee social responsibility? Is there a benefit-exchange motivation in addition to the identification motivation? To answer these questions, we take all A-share listed companies in China from 2006 to 2019 as our sample and measure senior executives' home bias based on whether a chairperson's or CEO's hometown is consistent with the firm's registration place. We use the score index for employee social responsibility from the CNRDS dataset to measure corresponding corporate fulfillment, and we systematically investigate the effects of senior executives' home bias on corporate employee social responsibility. The research methods detailed in this paper include theoretical analysis and empirical testing. We follow our empirical analysis with a series of robustness tests where propensity score matching (PSM) and the DID method are adopted. A cost–benefit analysis of motivations is also carried out. Finally, the heterogeneous effects of senior executives' home bias on employee social responsibility are analyzed via grouping regressions under different scenarios.

We use Chinese listed companies as our main research sample for two main reasons. First, affected by Confucianism and its traditional culture over several thousand years, China has developed into a country where there is a strong atmosphere of local favoritism [24]. There are many poems about homesickness in Chinese history. People's decisions and behavior are influenced by home bias, which is formed by their specific local favoritism. Second, in recent years, the Chinese government has continuously attached great importance to employee social responsibility and advocated building harmonious labor relations. In 2008, China's State-Owned Assets Supervision and Administration Commission (SASAC) issued the Guiding Opinions on Central Enterprises' Fulfillment of Social Responsibility. This authorized file emphasized that the most important topic of corporate social responsibility is employee social responsibility while specifying the associated responsibilities and how employees can develop themselves. Moreover, as enterprises are held responsible for all of society as well as the ecological environment through their fulfillment of employee social responsibility to achieve sustainable development. In July 2018, a meeting of the Political Bureau of CPC Central Committee regarded labor protection and improving social securities as a working focus. In November 2018, a conference of the National Labor Union on rights protection pointed out that institutional protection should be the main method of protecting employees' legal rights and building harmonious labor relations. Nevertheless, as mentioned above, Chinese enterprises still do not pay enough attention to employee social responsibility, responding to governmental policy inactively. The awareness of Chinese enterprises for fulfilling employee social responsibility thus urgently needs to be further improved.

In this paper, we demonstrate the following: First, the home bias of both CEO and chairperson can improve firms' fulfillment of employee social responsibility. Second, further cost-benefit analysis reveals that this result is due to not only identification but also benefit exchange. Although senior executives' home bias significantly decreases employee turnover rate, firms absorb more employment, significantly increasing their redundant personnel costs. Therefore, firms should balance the potential benefits and costs of home bias via trade-off. Third, in firms facing less market competition, firms with more governmental subsidies or state-owned firms, senior executives' home bias has a more significant promoting effect on the fulfillment of employee social responsibility, supporting the view of benefit exchange. The innovation and contribution of this paper are thus mainly reflected in the following: First, in the context of China's special culture, we incorporate home bias, which reflects personal features, into our

research framework of the potential motivations for employee social responsibility, filling a research gap in the literature on employee social responsibility. Second, in addition to supporting the identification motivation related to home bias, we creatively propose that senior executives' home bias promotes the fulfillment of employee social responsibility due to benefit exchange between enterprises and local governments. In this way, we extend the theoretical boundary of home bias research and enrich the literature on home bias. Third, this paper has a certain practical significance. Our findings enable relevant suggestions and have implications for solving the current problems with labor relations and other related issues in China. Furthermore, other emerging markets can refer to this research to obtain helpful insights for guiding and promoting enterprises to fulfill employee social responsibility.

## 2. Theoretical analysis and hypothesis

Employee social responsibility behavior requires firms to protect employees' legal rights and welfare and take ethical, economic and legal responsibility for them. Specifically, firms should provide employees with fair selection and promotion processes, reasonable welfare benefits, support for their education and development, and securities for health and safety and other related contents. Such responsibility is a very important component of corporate social responsibility [25]. According to stakeholder theory, as employees are a crucial interest party inside a firm, employee social responsibility is an important component of corporate social responsibility [26].

### 2.1. Identification

**2.1.1. Emotional identification.** Labor economics has already discussed the relationship between firms and employees. In addition to the information asymmetry between managers and shareholders, low communication frequency [27], community pressure [28], market competition [10] and other objective factors, the executive personal factor has also started to attract attention from scholars. Yonker [29] identifies U.S. senior executives' birthplace according to their social security number and finds that a CEO who works in his or her birthplace behaves friendlier toward employees. This finding is supported specifically by the fact that the layoff rate in these firms is 13% lower than in firms whose CEO is not from a firm-registered place when the economy is in a downturn. On the one hand, a firm's friendly attitude is a result of the extent to which the CEO is familiar with the local social network and the CEO's expectations for his or her retired life in this local community. On the other hand, a CEO might take into account that difficult layoffs could undermine social relationships with his or her friends or family members.

However, the identifying method used by Yonker [29] has a major drawback: all the places identified are in the U.S., while a social security number may not be consistent with a CEO's actual birthplace. In contrast, there is a strong sense of provincialism in China. Chinese home bias, influenced and developed by traditional culture, forms a natural emotional basis as well as a local social network between senior executives and their hometown. Identity economics theory [22] illustrates how individual utility can be maximized by an individual's focus on his or her own behavior, others' influence and his or her own sense of self-identity in groups. In China, a country filled with homeland love, one's hometown is an important characteristic of individual identification [18]. Due to their hometown identity and home bias, corporate senior executives tend to make biased decisions on issues related to their hometown. Therefore, we assume that in China, home bias provides senior executives with a strong sense of hometown identification; thus, these executives feel encouraged to fulfill employee social responsibility to repay their hometown.

According to Kong and Feng [30], senior executives who work in their hometown were born and raised in them. Therefore, they have strong feelings of both attachment and belonging to their hometown. The hometown identities of these senior executives motivate them to repay their hometown, and their common traits with their hometown and pleasing memories make them naturally kind toward their fellow hometown residents. Hence, repaying people from their hometown might be the most direct and effective way to repay their hometown. In China, even in the very distant past, there were examples where individuals cared for and helped their fellow hometown residents. For instance, Li [31] describes how Hairui, a famously incorruptible officer from the Ming Dynasty, due to his precious emotional identity with his hometown, cared much about its impoverished residents and fought for people of the Li nationality, who had been suppressed bloodily. Similarly, Cao et al. [18] find that a corporate CEO tends to set up a subsidiary firm in his or her hometown even if he or she is not working there to benefit the hometown's residents-repaying his or her hometown by promoting employment, raising incomes and so on. In firms that are registered in the hometown of their senior executives, local residents occupy a larger portion of employees [29]. Consequently, motivated to repay their hometown, senior executives who work in their hometown have more communications with employees and feel encouraged to ensure employees' benefits [27]. Accordingly, such senior executives pay more attention to employees' needs and benefits. Moreover, these senior executives also reduce their self-interested thoughts [24] and focus more on employees' benefits, treat their employees well, and actively fulfill employee social responsibility.

**2.1.2. Social network constraint.**   Identification theory is based on both the moral identities that individuals assign to groups and the social constraints groups place on individuals. Senior executives who work in their hometown are thus faced with more constraints via social comments. On the one hand, Chinese culture attaches great importance to thoughts such as "mutual benefits", "mutual help" and "upholding family honor" in a social relationship network. When senior executives work in their hometown, many of the corporate decisions they make will receive extensive attention from all local divisions. If senior executives behave in a way that depresses employees or damages employee benefits, they will be attacked not only by numerous social comments but also via condemnations from any of their relatives who believe such behaviors harm family honor, reflecting that these executives are unreasonable, ungrateful, and shameless [18]. In addition, both the residents and employees from their hometown might have higher expectations of local senior executives. As a result, once firms harm employees' benefits, local employees can magnify its negative effect through their social networks. On the other hand, when senior executives retire, they are highly likely to live in their hometown. Therefore, if they have not treated their employees well, they may be treated in an unfriendly or even hatful manner by the people around them. The effects of social pressure on senior executives' decisions on employee social responsibility have also been confirmed by relevant studies from outside China. Landier et al. [27] assert that senior executives care more about the benefits of their surrounding employees if they communicate frequently with these employees and that they carefully consider their own social reputation and personal image when making any relevant decision, as discharging surrounding employees or reducing their wages can embarrass these senior executives. Therefore, in such firms, when making decisions on lay-offs, corporate senior executives will prefer to dismiss employees who are distant from the corporate headquarters. Bassanini et al. [32] find that CEOs are reluctant to dismiss employees in corporate headquarters because they are concerned with damaging their reputation in the community where they work (usually, this communities is also where they live). Therefore, social comment constraints, triggered by identification, encourage senior executives to treat employees in a friendly manner and fulfill employee social responsibility actively.

## 2.2. Benefit exchange

Social exchange theory suggests that social exchange is a mutually beneficial activity where one party provides resources, assistance and support to another party and expects the latter's loyalty and repayment in return, even if these two parties are driven by different purposes [33, 34]. For senior executives who are attached to their hometown identity, despite the natural emotional reliance and social network pressure resulting from this identification, they and the firms they manage are still profit-driven. Consequently, senior executives and firms expect to gain equivalent profits in return for the costs and expenses incurred by actively performing employee social responsibility. In addition, senior executives' hometown social networks are more convenient and extensive than others, creating more opportunities for firms to communicate with governments, society, employees and other stakeholders when playing their interest games. The corporate fulfillment of employee social responsibility, on the one hand, can enhance employees' moral identification and promote their active organizational behavior [2]. On the other hand, such responsibility performance also helps governments execute policies promoting employment and labor security. Therefore, corporate senior executives' hometown identities can both encourage firms to fulfill more employee social responsibility and help them obtain the corresponding profits, as returns, when they play interest games with employees and governments.

In summary, the hometown identity of corporate senior executives can function in a personal, emotional way, out of social comment constraints to repay hometown, or through their social networks to help firms gain more "benefits". Regardless of how home bias works, it can induce senior executives to care more for employee benefits and thus to better fulfill employee social responsibility. Therefore, we posit the following:

H1: The home bias of senior executives significantly promotes the corporate fulfillment of employee social responsibility.

# 3. Research design

## 3.1. Sample selection and data resource

We select all A-share listed companies in China during 2006–2019 as our study sample. Furthermore, we modify the initial sample as follows: we eliminate financial institutions, insurance companies, ST and PT firms, and firms lacking available data. Finally, we obtain 2676 firm-year observations for the home bias of chairpersons and 2977 firm-year observations for the home bias of CEOs. Data on senior executives' hometown, birthplace and personal traits, such as gender, age and education, are obtained from the Wind and CSMAR databases, where resumes and information on the hometowns and birthplaces of senior executives (including chairpersons and CEOs) are comprehensively collected and processed. We also use Baidu, Sogou, Google and other search engines to address data deficiencies. Data on employee social responsibility are collected from detailed program data measuring whether firms have fulfilled employee social responsibility in the CNRDS database. Data on the degree of marketization and local legal environment are drawn from the China Marketization Index Report of Wang and Fan [35], and all other data come from the CSMAR database. Finally, to eliminate outlier influence, the sizes of all continuous variables are reduced by 1% from both the upper and lower extremes.

## 3.2. Definitions of variables

**3.2.1. Explained variables.** In this paper, employee social responsibility (EmpCsr) is the explained variable. The CNRDS database discloses detailed index data on whether firms fulfill

employee social responsibility. These data include, specifically, firm-level data on whether corporate employees hold shares in their firms, whether there are adequate employee benefits, whether firms adopt safety management systems and safety training, whether firms have occupational safety certification, professional training and superior communication channels for employees, and other indicators of firms' advantages. Each indicator is a binary variable that takes the value of 1 if a certain firm has the measure and 0 if the firm does not. Finally, we sum all indicators, add 1 to the sum value and then take its log form as the variable for employee social responsibility.

**3.2.2. Explanatory variables.** In this paper, the home bias of senior executives (Nati_Chair and Nati_CEO) is the core explanatory variable. While the basic moral characteristic of Chinese culture is relationship, hometown is an important dimension of any social relationship. Generally, people have a natural sense of attachment to their hometown, possibly leading them to feel a strong sense of identification and favoritism toward their hometown. Complying with the importance principle, we regard the chairperson or CEO of a certain company as our study objective and use the corresponding origin place to identify a hometown. While we assume that each senior executive has a sense of identity with his or her hometown, in reality, not every firm's senior executive works in his or her hometown. As a result, there are differences among different senior executives' feelings toward the places where their firms are registered [24]. If the place where a firm is registered is consistent with a senior executive's (chairperson's or CEO's) hometown, we assign this executive is the characteristic of home bias. Specifically, following Hu et al. [24] regarding the origin place of senior executives, we define the Nati_Chair value as 1 if the province where a firm is registered is consistent with the origin province of its chairperson and 0 otherwise. We also define the Nati_CEO value as 1 if the province where a firm is registered is consistent with the hometown province of its CEO and 0 otherwise.

**3.2.3. Control variables.** There are many factors that influence employee social responsibility. Following the literature [10], we define variables to control for the influences of employee social responsibility in four dimensions, namely, firm characteristics, corporate governance, macro levels and senior management traits. We use several specific indicators to measure each control variable. Firm characteristics are measured by firm size (Size), financial leverage (Lev), profitability (Roa) and cash flows from operating activities (Fcf). Variables of corporate governance include ownership structure (Stru) and proportion of the largest shareholder (Top1). Macro levels include local GDP per capita (PerGdp), degree of marketization (Market) and local legal environment (Law). Senior management traits include gender (Gender), age (Age), education (Degree). The specific definitions of these variables are shown in Table 1.

## 3.3. Model design

To examine the effects of the home bias of senior executives on the fulfillment of corporate employee social responsibility, the model we designed and examined is as follows:

$$EmpCsr = \alpha + \beta_1 Nati + \sum_i \beta_i Control_i + Year + Ind + \varepsilon \qquad (1)$$

In this equation, EmpCsr is the explained variable and represents the corporate fulfillment of employee social responsibility. Nati is the core explanatory variable and includes two variables: home bias of the chairperson (Nati_Chair) and home bias of the CEO (Nati_CEO). These two variables measure whether the chairperson or CEO has home bias for the place where the firm is registered. We mainly focus on β1: if β1 is significantly larger than 0, then

**Table 1. Variables and definitions.**

| Variable Trait | Variable Name | Variable Symbol | Variable Definition |
|---|---|---|---|
| Explained Variable | Employee social responsibility | EmpCsr | Whether corporate employees hold firms' own shares, whether there are excellent employee benefits, whether firms adopt safety management system and safety training, whether firms have occupational safety certification, professional training as well as superior communication channels for employees, and other indicators for firms' advantages. Each indicator takes the value of 1 if the firm has this characteristic and takes 0 otherwise. Finally, all these indicators are added, and the variable is measured by the sum's log value. |
| Explanatory Variable | Chairperson's home bias | Nati_Chair | Nati_Chair value is 1 if the province where the firm is registered is consistent with the province of its chairperson's hometown and 0 otherwise. |
| | CEO's home bias | Nati_CEO | Nati_CEO is 1 if the province where the firm is registered is consistent with the province of its CEO's hometown and 0 otherwise. |
| Control Variables | Firm size | Size | Natural logarithm of total assets at year end. |
| | Financial Leverage | Lev | Debt to asset ratio |
| | Profitability | Roa | Returns on total assets |
| | Cash flow from operating activities | Fcf | Net cash flows from operating activities, standardized by total assets at year end. |
| | Controlling shareholders | Top1 | Proportion of the largest shareholder |
| | Ownership structure | Stru | Concentration degree of five largest shareholders |
| | GDP per capital | PerGDP | Local total output divided by total population (in natural logarithm) |
| | Local legal environment | Law | Index for each local legal system, compiled by Wang and Fan [35] |
| | Degree of marketization | Market | Index for degree of each area's marketization, compiled by Wang and Fan [35] |
| | Chairperson's gender | Dender_Chair | Chairperson's gender |
| | | | Male = 1, Female = 0 |
| | Chairperson's age | Age_Chair | Chairperson's age |
| | Chairperson's education | Degree_Chair | Chairperson's education |
| | | | 1 = graduated from technical secondary school or less |
| | | | 2 = graduated from junior college |
| | | | 3 = undergraduate |
| | | | 4 = postgraduate |
| | | | 5 = doctorate |
| | CEO's gender | Dender_CEO | CEO's gender |
| | | | Male = 1, Female = 0 |
| | CEO's age | Age_CEO | CEO's age |
| | CEO's education | Degree_CEO | CEO's education |
| | | | 1 = graduated from technical secondary school or less |
| | | | 2 = graduated from junior college |
| | | | 3 = undergraduate |
| | | | 4 = postgraduate |
| | | | 5 = doctorate |

Hypothesis H1 is supported, i.e., the home bias of senior executives significantly promotes the fulfillment of corporate employee social responsibility.

## 4. Empirical results and analysis

### 4.1. Descriptive statistics

Table 2 shows the results for the descriptive statistics of the main variables. In Panel A, the mean, standard deviation, minimum and maximum for the variable EmpCsr are 1.619, 0.359, 0 and 2.197, respectively, illustrating the large individual differences with respect to the fulfillment of employee social responsibility among sampled firms, even after taking the logarithm for employee

**Table 2. Descriptive statistics.**

| Variables | Observations | Mean | Std. | Min | P25 | P50 | P75 | Max |
|---|---|---|---|---|---|---|---|---|
| **Panel A: Sample for chairperson home bias** | | | | | | | | |
| EmpCsr | 2676 | 1.619 | 0.359 | 0 | 1.386 | 1.609 | 1.792 | 2.197 |
| Nati_Chair | 2676 | 0.677 | 0.468 | 0 | 0 | 1 | 1 | 1 |
| Size | 2676 | 22.67 | 1.333 | 18.97 | 21.70 | 22.57 | 23.55 | 26.60 |
| Lev | 2676 | 0.452 | 0.195 | 0.0540 | 0.303 | 0.459 | 0.604 | 1.374 |
| Roa | 2676 | 0.0550 | 0.0570 | -0.412 | 0.0240 | 0.0490 | 0.0830 | 0.205 |
| Fcf | 2676 | 0.0610 | 0.0760 | -0.212 | 0.0180 | 0.0590 | 0.104 | 0.264 |
| Top1 | 2676 | 0.364 | 0.155 | 0.0880 | 0.238 | 0.344 | 0.486 | 0.750 |
| Stru | 2676 | 0.546 | 0.158 | 0.192 | 0.428 | 0.547 | 0.662 | 0.894 |
| PerGDP | 2676 | 10.89 | 0.567 | 8.972 | 10.49 | 10.95 | 11.36 | 11.97 |
| Law | 2676 | 5.419 | 2.992 | -1.130 | 3.190 | 5.040 | 7.270 | 12.75 |
| Market | 2676 | 7.754 | 1.749 | 2.870 | 6.620 | 7.800 | 9.260 | 11.04 |
| Gender_Chair | 2676 | 0.925 | 0.264 | 0 | 1 | 1 | 1 | 1 |
| Age_Chair | 2676 | 53.15 | 7.002 | 24 | 48 | 53 | 57 | 79 |
| Degree_Chair | 2676 | 3.563 | 0.918 | 1 | 3 | 4 | 4 | 5 |
| **Panel B: Sample for CEO home bias** | | | | | | | | |
| EmpCsr | 2977 | 1.616 | 0.339 | 0 | 1.386 | 1.609 | 1.792 | 2.197 |
| Nati_CEO | 2977 | 0.608 | 0.488 | 0 | 0 | 1 | 1 | 1 |
| Size | 2977 | 22.75 | 1.458 | 18.97 | 21.70 | 22.60 | 23.64 | 26.60 |
| Lev | 2977 | 0.462 | 0.204 | 0.0540 | 0.307 | 0.473 | 0.620 | 1.374 |
| Roa | 2977 | 0.0540 | 0.0550 | -0.412 | 0.0220 | 0.0460 | 0.0820 | 0.205 |
| Fcf | 2977 | 0.0580 | 0.0760 | -0.212 | 0.0140 | 0.0560 | 0.103 | 0.264 |
| Top1 | 2977 | 0.382 | 0.156 | 0.0880 | 0.252 | 0.377 | 0.503 | 0.750 |
| Stru | 2977 | 0.561 | 0.159 | 0.192 | 0.447 | 0.557 | 0.675 | 0.894 |
| PerGDP | 2977 | 10.84 | 0.555 | 8.972 | 10.45 | 10.90 | 11.24 | 11.97 |
| Law | 2977 | 5.312 | 2.867 | -1.130 | 3.230 | 5.040 | 7.190 | 12.75 |
| Market | 2977 | 7.669 | 1.713 | 2.870 | 6.530 | 7.730 | 9.150 | 11.04 |
| Gender_CEO | 2977 | 0.942 | 0.235 | 0 | 1 | 1 | 1 | 1 |
| Age_CEO | 2977 | 48.98 | 6.233 | 27 | 45 | 49 | 53 | 75 |
| Degree_CEO | 2977 | 3.648 | 0.806 | 1 | 3 | 4 | 4 | 5 |

social responsibility. In Panel B, descriptive statistics of the variable EmpCsr have a similar feature. Nati_Chair and Nati_CEO, variables representing whether the place where a listed firm is registered is consistent with its senior executive's hometown, have an average of 0.677 and 0.608, respectively. These averages show that 67.7% of the listed firms we have sampled hire a chairperson whose hometown is the same as the firm-registered place and that 60.8% of listed firms hire a CEO of this type. In other words, in our sample, 67.7% of firm chairpersons have home bias and 60.8% of firm CEOs have home bias. The averages for the variables Gender_Chair, Age_Chair and Degree_Chair are 0.925, 53.15 and 3.563, respectively, indicating that the chairpersons in the sampled firms are mostly males, are aged 53 years on average, and have an average education level of undergraduate or above. Regarding the personal characteristics of the CEOs, their average age is lower, while their gender and education show similar features.

## 4.2. Basic regression results

Table 3 shows the regression results between corporate employee social responsibility and the home bias of senior executives. The effects of chairperson home bias are shown in Columns

**Table 3. Home bias of senior executives and employee social responsibility.**

| | Chairperson | | | CEO | | |
|---|---|---|---|---|---|---|
| | (1) | (2) | (3) | (4) | (5) | (6) |
| | EmpCsr | EmpCsr | EmpCsr | EmpCsr | EmpCsr | EmpCsr |
| Nati_Chair | 0.030** | 0.029** | 0.027* | | | |
| | (2.25) | (2.09) | (1.95) | | | |
| Nati_CEO | | | | 0.032*** | 0.036*** | 0.036*** |
| | | | | (2.73) | (2.86) | (2.83) |
| Size | 0.033*** | 0.033*** | 0.032*** | 0.043*** | 0.043*** | 0.042*** |
| | (5.24) | (5.22) | (4.91) | (7.67) | (7.66) | (7.26) |
| Lev | -0.061 | -0.058 | -0.055 | -0.057 | -0.055 | -0.054 |
| | (-1.22) | (-1.18) | (-1.10) | (-1.33) | (-1.28) | (-1.24) |
| Roa | 0.108 | 0.088 | 0.089 | -0.010 | -0.021 | -0.012 |
| | (0.68) | (0.55) | (0.55) | (-0.07) | (-0.16) | (-0.09) |
| Fcf | -0.012 | -0.017 | -0.030 | -0.007 | -0.001 | -0.005 |
| | (-0.13) | (-0.18) | (-0.32) | (-0.09) | (-0.01) | (-0.05) |
| Top1 | -0.125** | -0.117* | -0.128** | -0.017 | -0.008 | -0.013 |
| | (-2.05) | (-1.91) | (-2.07) | (-0.33) | (-0.15) | (-0.25) |
| Stru | 0.074 | 0.070 | 0.084 | -0.010 | -0.021 | -0.016 |
| | (1.27) | (1.20) | (1.43) | (-0.20) | (-0.41) | (-0.31) |
| PerGDP | | -0.063** | -0.063** | | -0.010 | -0.010 |
| | | (-2.24) | (-2.26) | | (-0.39) | (-0.40) |
| Law | | 0.004 | 0.005 | | 0.006 | 0.006 |
| | | (1.08) | (1.13) | | (1.51) | (1.55) |
| Market | | 0.010 | 0.010 | | 0.003 | 0.003 |
| | | (1.36) | (1.35) | | (0.45) | (0.38) |
| Gender_Chair (Gender_CEO) | | | 0.006 | | | 0.027 |
| | | | (0.25) | | | (1.02) |
| Age_Chair (Age_CEO) | | | 0.002* | | | 0.000 |
| | | | (1.69) | | | (0.19) |
| Degree_Chair (Degree_CEO) | | | -0.001 | | | -0.001 |
| | | | (-0.08) | | | (-0.11) |
| Constant | 0.499*** | 1.245*** | 1.199*** | 0.296** | 0.334 | 0.318 |
| | (3.56) | (4.11) | (3.91) | (2.32) | (1.43) | (1.35) |
| Ind&Year | Yes | Yes | Yes | Yes | Yes | Yes |
| Observations | 2676 | 2676 | 2676 | 2977 | 2977 | 2977 |
| R-squared | 0.2448 | 0.2466 | 0.2474 | 0.2084 | 0.2107 | 0.2110 |

Notes: Robust t-statistic is reported in parentheses.

***, ** and * indicate the significance at the level of 1%, 5% and 10%, respectively.

(1)-(3). In Column (1), only firm characteristics and corporate governance are controlled for, and thus Nati_Chair is positive and significant at the 5% level. In Column (2), macro levels such as GDP per capita (PerGDP), local legal environment (Law), degree of marketization (Market) and other variables are controlled for, in addition to firm characteristics and corporate governance. Nati_Chair is therefore still positive and significant at the 5% level. Column (3) further controls for the variables of chairpersons' personal characteristics, such as gender (Gender_Chair), age (Age_Chair), and educational background (Degree_Chair). Hence, Nati_Chair is significantly positive at the 10% level. Columns (4)-(6) focus on CEOs' home

bias. Nati_CEO values in Columns (4)-(6) are all positive and significant at the 1% level. Moreover, the coefficients in Columns (4)-(6) are slightly larger than the corresponding coefficients in Columns (1)-(3). In summary, gradually controlling for firm characteristics, corporate governance, macrolevel variables and senior executives' personal traits, we find a significant positive relationship between the home bias of chairpersons or CEOs and the fulfillment of corporate social responsibility. Therefore, H1 is supported: A chairperson's or CEO's home bias helps promote the fulfillment of corporate employee social responsibility, and the effects of the chairperson sample and CEO sample are nearly equivalent. The results in Table 3 affirm Hypothesis H1.

Zhu et al. [36] find that local senior executives influence enterprises to fulfill social responsibility. Table 3 further proves Zhu et al.'s finding by showing our results regarding employee social responsibility, a subsidiary social responsibility. Although our findings show that the influence of home bias uses the channel of identification [20, 37], we do not think home bias functions only because of emotional identification (senior executives want to reward their hometown) [38] or hometown social network constraints [18]. Hejjas et al. [39] show that a complex combination of public needs, governmental expectations and other reasons encourages an increasing number of enterprises to fulfill social responsibility. To some extent, Hejjas et al.'s finding also reflects the complexity of enterprises' motivations to fulfill social responsibility, an important one of which is gaining benefits and resources [40]. We suggest that the identity of fellow hometown residents gives a senior executive a wider and more convenient social network and enables him or her to play an interest game with employees or the relevant government to obtain corresponding benefits based on the premise of fulfilling social responsibility. Therefore, the conclusions we have obtained might be due to benefit exchange, which has never been discussed in the literature. Via further analysis, we thus deeply discuss how senior executives' home bias promotes the fulfillment of employee social responsibility due to benefit-exchange motivation.

## 4.3. Robustness test

**4.3.1. Propensity score matching.** We adopt propensity score matching to reduce the effects of potential endogenous problems.

First, we group listed firms that are exposed to home bias of senior executives as the experimental group and those without home bias as the control group. Second, we conduct logit regression with variables including firm size (Size), asset-liability ratio (Lev), profitability (Roa), cash flows from operating activities (Fcf), degree of marketization (Market), local legal environment (Law), gender (Gender), age (Age) and education (Degree) as covariates in this regression and compute propensity scores. Finally, each control group sample was matched with each experimental group using the nearest-neighbor method without being returned and with a caliper range of 0.001. We obtain 1150 matched samples (575 couples) for the sample of chairperson home bias and 1364 matched samples (682 couples) for the sample of CEO home bias.

Table 4 displays the Group t test for matching samples. It can be concluded from the table that there is no significant difference in the grouped samples with regard to all explanatory variables after the samples are matched. Matched samples are then regressed in Model (1). Table 5 displays results based on the propensity score matching method. Clearly, these results comply with initial conclusions, as Nati_Chair and Nati_CEO are positive and significant at the 1% and 5% levels, respectively.

**4.3.2. Impacts of senior executive turnover.** Existing home bias might be eliminated by senior executive turnover in listed firms. This exogenous event provides us with a new way to

**Table 4. T test for the experimental group and control group.**

| | Chairperson | | | CEO | | |
|---|---|---|---|---|---|---|
| | Experimental group | Control group | T-statistic | Experimental group | Control group | T-statistic |
| Size | 22.825 | 22.798 | 0.42 | 22.767 | 22.704 | 0.92 |
| Lev | 0.465 | 0.467 | -0.23 | 0.462 | 0.457 | 0.49 |
| Roa | 0.057 | 0.055 | 0.7 | 0.054 | 0.055 | -0.16 |
| Fcf | 0.061 | 0.057 | 0.99 | 0.058 | 0.058 | 0.02 |
| Top1 | 0.367 | 0.364 | 0.38 | 0.386 | 0.374 | 1.56 |
| Stru | 0.551 | 0.545 | 0.77 | 0.561 | 0.552 | 1.07 |
| PerGDP | 10.873 | 10.870 | 0.11 | 10.845 | 10.825 | 0.76 |
| Law | 5.462 | 5.404 | 0.42 | 5.287 | 5.196 | 0.65 |
| Market | 7.653 | 7.698 | -0.55 | 7.578 | 7.589 | 0.77 |
| Gender_Chair (Gender_CEO) | 0.954 | 0.957 | -0.29 | 0.946 | 0.937 | 0.77 |
| Age_Chair(age_CEO) | 52.756 | 52.872 | -0.37 | 49.318 | 49.021 | 0.97 |
| Degree_Chair(Degree_ Chair) | 3.700 | 3.667 | 0.85 | 3.658 | 3.665 | -0.20 |

deal with potential endogeneity. While the home bias of senior executives in listed companies can promote the fulfillment of corporate employee responsibility, such performance is likely to decline when such home bias disappears due to top management turnover [41]. Therefore, from the initial sample of listed companies with home bias of senior executives, we select listed companies where home bias initially existed but has recently disappeared because of top management turnover to eliminate the interference of other factors. We define the variable Change_Chair (Change_CEO) to measure whether the firm-registered place is still consistent with the chairperson's (CEO's) origin place after top management turnover. Change_Chair (Change_CEO) takes the value of 1 if the two places are still the same and 0 otherwise. Moreover, we build a difference-in-difference model with multiple time points to identify how the fulfillment of corporate employee social responsibility is impacted when the home bias of senior executives disappears.

$$EmpCsr = \alpha + \beta_1 Change + \sum_i \beta_i Control_i + Year + Ind + \varepsilon \tag{2}$$

In the above equation, the variable Change includes Change_Chair, a variable measuring the effect of the disappearance of the chairperson's home bias, and Change_CEO, a variable measuring the effect of the disappearance of the CEO's home bias. The control variables used in Model (2) are the same as those used in Model (1). Table 6 displays these results. Here, both Change_Chair and Change_CEO are not significant and become negative, illustrating that the promotion effect of home bias on fulfilling employee social responsibility will become unapparent and even suppressive when listed firms experience the disappearance of an existing home bias. In this way, we demonstrate causality between the home bias of senior executives and corporate employee social responsibility from a contradictory perspective.

**4.3.3. Sensitivity test for indicators of home bias.** First, according to Hu et el. [24], to measure the home bias of senior executives, we use both the province scale and city scale to obtain comprehensive matching results for hometown and firm-registered places. In the previous analysis, we used provinces to identify senior executives' origin place and compare it with the province where their firm is registered. We choose the provincial level because we find that most information about senior executives' hometown is often disclosed at the provincial level and rarely at the city level. However, it seems lack sufficient precision to simply replace specific

**Table 5. OLS regression based on propensity score matching.**

|  | Chairperson | CEO |
|---|---|---|
|  | EmpCsr | EmpCsr |
| Nati_Chair | 0.052*** |  |
|  | (2.62) |  |
| Nati_CEO |  | 0.039** |
|  |  | (2.31) |
| Size | 0.033*** | 0.034*** |
|  | (3.33) | (3.56) |
| Lev | -0.109 | 0.024 |
|  | (-1.49) | (0.34) |
| Roa | 0.189 | 0.296 |
|  | (0.90) | (1.32) |
| Fcf | 0.053 | 0.130 |
|  | (0.38) | (0.97) |
| Top1 | -0.119 | 0.080 |
|  | (-1.32) | (0.98) |
| Stru | 0.047 | -0.117 |
|  | (0.53) | (-1.40) |
| PerGDP | -0.100** | -0.007 |
|  | (-2.39) | (-0.19) |
| Law | 0.011* | -0.001 |
|  | (1.82) | (-0.21) |
| Market | 0.005 | 0.019* |
|  | (0.43) | (1.76) |
| Gender_Chair (Gender_CEO) | 0.021 | 0.036 |
|  | (0.58) | (0.87) |
| Age_Chair (Age_CEO) | 0.000 | -0.000 |
|  | (0.23) | (-0.14) |
| Degree_Chair (Degree_CEO) | 0.015 | -0.019* |
|  | (1.25) | (-1.67) |
| Constant | 1.388*** | 0.455 |
|  | (3.43) | (1.08) |
| Ind&Year | Yes | Yes |
| Observations | 1150 | 1364 |
| R-squared | 0.2974 | 0.2451 |

Notes: Robust t-statistic is reported in parentheses.

***, ** and * indicate the significance at the level of 1%, 5% and 10%, respectively.

hometown places with provinces. Consequently, we use province or city as to match senior executives' origin place with firm-registered place to measure home bias.

Specifically, if the information about the hometown of senior executives can be specified at the city level, then we can compare whether the origin city is the same as the firm-registered city. City_Chair or City_CEO takes the value of 1 if the two cities are consistent and 0 otherwise. For samples that cannot be shown at the city level, we match hometown place and firm-registered place on the provincial basis. In this way, the mechanism through which City_Chair or City_CEO takes a value of 1 or 0 is the same as above, but the standard becomes provincial again. Model (1) is regressed by including the two variables City_Chair and City_CEO. The

**Table 6. Impacts of senior executive turnover.**

|  | Chairperson | CEO |
|---|---|---|
|  | EmpCsr | EmpCsr |
| Change_Chair | -0.059 |  |
|  | (-0.93) |  |
| Change_CEO |  | -0.024 |
|  |  | (-0.46) |
| Size | 0.023 | 0.006 |
|  | (1.12) | (0.25) |
| Lev | -0.071 | -0.051 |
|  | (-0.47) | (-0.27) |
| Roa | -0.357 | -1.017** |
|  | (-0.93) | (-2.29) |
| Fcf | -0.527** | -0.175 |
|  | (-1.99) | (-0.65) |
| Top1 | -0.268 | -0.285 |
|  | (-1.31) | (-1.30) |
| Stru | 0.400** | 0.218 |
|  | (2.08) | (1.03) |
| PerGDP | -0.160 | -0.153 |
|  | (-1.39) | (-1.46) |
| Law | 0.015 | -0.009 |
|  | (1.33) | (-0.69) |
| Market | -0.000 | 0.084** |
|  | (-0.01) | (2.12) |
| Gender_Chair (Gender_CEO) | -0.057 | -0.155* |
|  | (-0.63) | (-1.94) |
| Age_Chair (Age_CEO) | 0.001 | -0.003 |
|  | (0.17) | (-0.77) |
| Degree_Chair (Degree_CEO) | -0.013 | 0.038 |
|  | (-0.41) | (1.32) |
| Constant | 2.857** | 2.308** |
|  | (2.50) | (2.39) |
| Ind&Year | Yes | Yes |
| Observations | 294 | 313 |
| R-squared | 0.2585 | 0.2985 |

Notes: Robust t-statistic is reported in parentheses.

***, ** and * indicate the significance at the level of 1%, 5% and 10%, respectively.

results of this novel regression are shown in the first two columns in Table 7. Here, City_Chair and City_CEO are both positive and significant at the 10% level, in accordance with our initial conclusion.

Second, although hometown usually coincides with birthplace, it is not rare that the two places are consistent. Senior executives in listed firms might have a stronger sense of home-town identification with their birthplace. Therefore, to mitigate the potential weakness of our assumption, we adopt a substitutive indicator that measures the home bias of senior executives by checking whether their birth province is the same as the firm-registered province, and then we run the regression again.

**Table 7. Robustness test: Sensitivity test.**

| | Origin Place at Provincial or City Level | | Province of Birthplace | | Negative Employee Social Responsibility | |
|---|---|---|---|---|---|---|
| | **Chairperson** | **CEO** | **Chairperson** | **CEO** | **Chairperson** | **CEO** |
| | **EmpCsr** | **EmpCsr** | **EmpCsr** | **EmpCsr** | *EmpBad* | *EmpBad* |
| City_Chair | 0.022* | | | | | |
| | (1.77) | | | | | |
| City_CEO | | 0.022* | | | | |
| | | (1.71) | | | | |
| Birt_Chair | | | 0.024 | | | |
| | | | (1.63) | | | |
| Birt_CEO | | | | 0.039* | | |
| | | | | (1.89) | | |
| Nati_Chair | | | | | -0.011** | |
| | | | | | (-2.31) | |
| Nati_CEO | | | | | | -0.010** |
| | | | | | | (-1.97) |
| Size | 0.032*** | 0.035*** | 0.040*** | 0.029*** | 0.020*** | 0.021*** |
| | (5.08) | (5.29) | (5.46) | (3.19) | (7.11) | (6.70) |
| Lev | -0.052 | -0.097** | -0.102* | -0.025 | 0.019 | 0.016 |
| | (-1.09) | (-2.04) | (-1.89) | (-0.36) | (1.36) | (1.17) |
| Roa | 0.094 | -0.017 | -0.051 | 0.113 | -0.070 | -0.098 |
| | (0.60) | (-0.11) | (-0.25) | (0.60) | (-1.20) | (-1.41) |
| Fcf | -0.026 | -0.001 | -0.058 | 0.114 | 0.015 | 0.025 |
| | (-0.28) | (-0.01) | (-0.60) | (0.95) | (0.49) | (0.77) |
| Top1 | -0.112* | -0.121** | -0.151** | -0.031 | -0.089*** | -0.084*** |
| | (-1.85) | (-1.97) | (-2.22) | (-0.35) | (-4.03) | (-3.90) |
| Stru | 0.087 | 0.034 | 0.088 | -0.071 | 0.080*** | 0.077*** |
| | (1.47) | (0.56) | (1.34) | (-0.75) | (3.61) | (3.32) |
| PerGDP | -0.069** | -0.047* | -0.054* | -0.052 | -0.029*** | -0.026*** |
| | (-2.49) | (-1.68) | (-1.83) | (-1.26) | (-3.30) | (-2.76) |
| Law | 0.004 | 0.005 | 0.011** | 0.003 | -0.001 | -0.000 |
| | (1.04) | (1.21) | (2.25) | (0.45) | (-0.63) | (-0.29) |
| Market | 0.012 | 0.006 | -0.004 | 0.015 | 0.007*** | 0.007*** |
| | (1.54) | (0.79) | (-0.46) | (1.24) | (3.31) | (2.81) |
| Gender_Chair (Gender_CEO) | 0.010 | 0.008 | -0.037 | 0.061 | -0.004 | -0.006 |
| | (0.39) | (0.30) | (-1.04) | (1.60) | (-0.29) | (-0.43) |
| Age_Chair (Age_CEO) | 0.001 | 0.001 | -0.000 | 0.003* | 0.000 | 0.000 |
| | (1.61) | (1.17) | (-0.11) | (1.81) | (0.71) | (0.61) |
| Degree_Chair (Degree_CEO) | -0.002 | -0.007 | 0.001 | -0.000 | -0.003 | -0.000 |
| | (-0.24) | (-1.00) | (0.10) | (-0.00) | (-1.36) | (-0.17) |
| Constant | 1.053*** | 0.874*** | 1.231*** | 1.214*** | -0.211** | -0.248*** |
| | (3.86) | (3.19) | (3.71) | (2.80) | (-2.36) | (-2.60) |
| Ind&Year | Yes | Yes | Yes | Yes | Yes | Yes |
| Observations | 2722 | 2507 | 2205 | s1322 | 3288 | 2987 |
| R-squared | 0.2481 | 0.2282 | 0.2552 | 0.2533 | 0.0671 | 0.0654 |

Notes: Robust t-statistic is reported in parentheses.

***, ** and * indicate the significance at the level of 1%, 5% and 10%, respectively.

The two columns in the middle of Table 7 show these results. Here, both Birt_Chair and Birt_CEO are positive and Birt_CEO is significant at the 10% level, clearly supporting our initial conclusion.

**4.3.4. Sensitivity test for indicators of corporate employee social responsibility.** The CNRDS database discloses data on whether firms are involved in employee safety disputes and whether they have dismissed vast numbers of employees. According to this database, EmpBad is defined as the indicator for negative employee social responsibility and takes the value of 1 if there is an employee safety dispute or lay-off and 0 otherwise. Model (1) is regressed with this newly defined variable and returns the results shown in the last two columns in Table 7. Here, Nati_Chair and Nati_CEO are both negative and significant at the 5% level. Therefore, both the home bias of chairpersons and that of CEOs reduce the slack performance of corporate employee social responsibility, reducing safety disputes between firms and employees as well as lay-off issues. Our initial conclusion is thus justified from a contradictory perspective.

**4.3.5. Controlling for the home bias of chairpersons.** Since the chairperson is the highest leader and decision-maker in a firm, the strategic behavior of the CEO is inevitably affected by the chairperson. As a result, the promotion effect of the CEO's home bias on corporate employee social responsibility might be influenced by the home bias of the chairperson, and ignoring such a possible influence will cause estimation biases and errors. Therefore, when regressing the CEO's home bias on corporate employee social responsibility, we control for Nati_Chair (variable of chairperson home bias) to eliminate the possible influences of the chairperson's home bias.

These regression results are shown in Table 8. When we control for chairperson home bias and gradually include control variables (firm characteristics, corporate governance, macrolevel indicators and CEO personal traits), Nati_CEO in different experimental cases is positive and significant at the 5% level. This result further proves that CEO home bias has a promotion effect on the fulfillment of corporate employee social responsibility, consistent with our original conclusion.

# 5. Further analysis

## 5.1. Benefit effect, triggered by senior executives' home bias, promoting the fulfillment of employee social responsibility (firm-employee)

Among all relevant stakeholders, corporate employees benefit directly from the corporate fulfillment of employee social responsibility. In firms where the corporate fulfillment of employee social responsibility is better, there are more reasonable wage levels, more comfortable working conditions, more training opportunities for professional development and more attention given to health and safety. According to social exchange theory, there is a mutually beneficial relationship between firms and employees. Specifically, firms provide their employees with resources, benefits and support in exchange for their loyalty and repayment. Flammer and Luo [42] point out that one strategic tool for firms to enhance employee loyalty is fulfilling corporate social responsibility. Through endeavors related to corporate social responsibility and felt by employees, firms can effectively reduce employees' intentions to resign, enhance their working performance [43] and encourage them to actively perform organizational citizenship behavior [2]. Therefore, we infer that corporate home bias can not only encourage firms to perform employee social responsibility but also strengthen the stability of their corporate staff. In our quantitative model, this increased stability is measured by a lower employee turnover rate.

To justify the inferences above, we refer to Wei [44] and build the employee turnover rate indicator, i.e., the difference between last-year employee number and current-year, divided by

**Table 8. Control home bias of chairperson.**

| | CEO | | |
|---|---|---|---|
| | EmpCsr | EmpCsr | EmpCsr |
| Nati_CEO | 0.049** | 0.050** | 0.050** |
| | (1.99) | (2.02) | (2.03) |
| Size | 0.044*** | 0.044*** | 0.044*** |
| | (7.62) | (7.67) | (7.67) |
| Lev | -0.053 | -0.051 | -0.053 |
| | (-1.19) | (-1.14) | (-1.19) |
| Roa | -0.077 | -0.090 | -0.077 |
| | (-0.55) | (-0.63) | (-0.54) |
| Fcf | -0.045 | -0.043 | -0.044 |
| | (-0.49) | (-0.47) | (-0.49) |
| Top1 | -0.007 | 0.002 | 0.000 |
| | (-0.14) | (0.04) | (0.00) |
| Stru | -0.002 | -0.009 | -0.008 |
| | (-0.03) | (-0.17) | (-0.14) |
| PerGDP | | -0.027 | -0.027 |
| | | (-1.01) | (-1.01) |
| Law | | 0.005 | 0.005 |
| | | (1.20) | (1.18) |
| Market | | 0.006 | 0.007 |
| | | (0.80) | (0.83) |
| Gender_CEO | | | 0.023 |
| | | | (0.85) |
| Age_CEO | | | -0.000 |
| | | | (-0.40) |
| Degree_CEO | | | 0.004 |
| | | | (0.51) |
| *Nati_Chair* | -0.017 | -0.016 | -0.016 |
| | (-0.68) | (-0.67) | (-0.66) |
| Constant | 0.290** | 0.789*** | 0.783*** |
| | (2.23) | (2.88) | (2.80) |
| Ind&Year | Yes | Yes | Yes |
| Observations | 2772 | 2772 | 2772 |
| R-squared | 0.2048 | 0.2062 | 0.2066 |

Notes: Robust t-statistic is reported in parentheses.

***, ** and * indicate the significance at the level of 1%, 5% and 10%, respectively.

the average of these two years' employee numbers. The higher this indicator is, the higher the employee turnover rate.

The regression results are reported in Table 9. Clearly, here, Nati_Chair and Nati_CEO are negative and significant at the 5% and 1% levels, respectively. This shows that senior executives' home bias can both promote the fulfillment of corporate employee social responsibility and significantly lower employment turnover rate. Generally, employees will gain a sense of identity when they perceive the corporate fulfillment of social responsibility [45, 46]. Their sense of identity with their enterprise reflects that employees are proud of and loyal to their enterprise [47]. The results in Table 9 further prove that when senior executives' home bias promotes enterprises to better fulfill employee social responsibility, employees are better able

**Table 9. Home bias of senior executives and employee turnover rate.**

|  | EmpLoss | |
|---|---|---|
|  | **Chairperson** | **CEO** |
| Nati_Chair | -0.010** |  |
|  | (-2.02) |  |
| Nati_CEO |  | -0.012*** |
|  |  | (-2.70) |
| Size | -0.004 | -0.004 |
|  | (-1.57) | (-1.61) |
| Lev | -0.007 | 0.000 |
|  | (-0.36) | (0.00) |
| Roa | -0.235*** | -0.268*** |
|  | (-3.15) | (-3.60) |
| Fcf | 0.022 | 0.043 |
|  | (0.54) | (1.12) |
| Top1 | -0.009 | -0.012 |
|  | (-0.40) | (-0.63) |
| Stru | -0.016 | -0.021 |
|  | (-0.83) | (-1.19) |
| PerGDP | 0.024** | 0.020** |
|  | (2.04) | (2.15) |
| Law | 0.001 | 0.001 |
|  | (0.91) | (0.87) |
| Market | -0.007* | -0.006* |
|  | (-1.95) | (-1.87) |
| Gender_Chair (Gender_CEO) | 0.015** | 0.003 |
|  | (2.20) | (0.54) |
| Age_Chair (Age_CEO) | 0.000 | 0.000 |
|  | (1.05) | (0.43) |
| Degree_Chair (Degree_CEO) | -0.002 | -0.003 |
|  | (-0.92) | (-0.89) |
| Constant | -0.028 | 0.025 |
|  | (-0.25) | (0.28) |
| Ind &Year | Yes | Yes |
| Observations | 2646 | 2904 |
| R-squared | 0.0424 | 0.0394 |

Notes: Robust t-statistic is reported in parentheses.

***, ** and * indicate the significance at the level of 1%, 5% and 10%, respectively.

to perceive this kind of improvement themselves, to develop a stronger sense of loyalty and to be less likely to resign. Consequently, enterprises can build a more stable team of employees, which fosters innovative corporate development [48]. This advantage is one of the benefits of promoting the effect of senior executives' home bias on fulfilling employee social responsibility.

## 5.2. Costs of senior executives' home bias when promoting the fulfillment of employee social responsibility (firm-government)

In addition to employees, a party that can benefit from the corporate fulfillment of employee social responsibility is a government. Employment is the most important aspect of people's

livelihood and an essential basis for the sustainable development of the economy [49]. The political bureau of the CPC central committee has emphasized that governments should help firms stabilize internal position changes and promote social employment stability. Once central advocates and commands are assigned to a local government, they become political missions that local governments should undertake and complete. On the one hand, because administration is quite important for corporate operational decisions [50], corporate behavior that is consistent with governmental goals helps firms develop institutional legitimacy and cultivate a more favorable external operating environment [51, 52]. From this perspective, the home identity of a senior executive makes it easier to build a social network with the local government, helping the firm follow the "stabilizing employment" public governance goal more accurately. On the other hand, for the local government, senior executives with a hometown identity tend to be focused on policies. The government can communicate with firms through local entrepreneurs, convey its goal of social stability to firms and influence them with this goal. In this way, a government can both promote economic development and reach the public goal of social stability. In summary, we infer that senior executives' home bias not only promotes the corporate fulfillment of employee social responsibility but also causes firms to take on more employment tasks, which are specified by a higher level of staff redundancy.

To demonstrate this inference, we build the indicator for corporate level staff redundancy [53]:

$$Y = \alpha + \beta Size + \theta Captial + \omega Growth + \varepsilon \qquad (3)$$

In the above model, Y is the scale of employees and is computed by dividing the number of employees by total assets and then multiplying by 1,000,000. Size is the natural logarithm of total assets. Capital is the degree of capital intensity and is measured by dividing fixed assets by total assets. Growth represents corporate growth potential and is measured by the growth rate of operating incomes. After defining these variables, Model (3) is regressed based on different industries and different years, and residuals obtained represent levels of corporate redundant employees (Burden).

The regression results are shown in Table 10. Here, Nati_Chair and Nati_CEO are both positive and significant at the 5% level. Senior executives' home bias thus not only promotes firms to perform employee social responsibility but also makes them take on more employment tasks, leading to a higher level of staff redundancy. Enterprises usually regard the fulfillment of employee social responsibility and other internal responsibilities as cost-incurring behaviors [3]. Our results in Table 10 show that while senior executives' home bias promotes the fulfillment of employee social responsibility, it also produces a higher level of redundant personnel. Accordingly, such enterprises accept a certain level of political burden [54], undertake loads of employment tasks [54], and incur additional costs [55].

In summary, benefit-cost analysis reveals that the significant promoting effect of home bias on fulfilling employee social responsibility stems mostly from benefit exchange between senior executives and enterprises, governments or employees. Senior executives' fulfillment of employee social responsibility can only be encouraged and promoted with vast benefits such as rewards. This condition is clearly shown in the next section's heterogeneity analysis.

## 6. Heterogeneity analysis

In the previous section, we have analyzed the corresponding corporate benefits and costs of senior executives' fulfillment of employee social responsibility in their hometown. Here, we perform heterogeneity analysis to further discuss how firms engage in trade-off regarding their economic profits when performing employee social responsibility in their hometown, indirectly verifying the benefit exchange between firms and their stakeholders.

**Table 10. Senior executive home bias and employee redundancy.**

| | Burden | |
| --- | --- | --- |
| | Chairperson | CEO |
| Nati_Chair | 0.058** | |
| | (2.07) | |
| Nati_CEO | | 0.056** |
| | | (2.41) |
| Size | -0.119*** | -0.119*** |
| | (-7.40) | (-7.69) |
| Lev | -0.009 | 0.084 |
| | (-0.08) | (0.81) |
| Roa | -0.106 | 0.008 |
| | (-0.53) | (0.04) |
| Fcf | 0.509** | 0.434** |
| | (2.47) | (2.19) |
| Top1 | 0.049 | 0.056 |
| | (0.36) | (0.42) |
| Stru | 0.330*** | 0.293*** |
| | (3.43) | (3.11) |
| PerGDP | -0.161*** | -0.245*** |
| | (-2.75) | (-4.55) |
| Law | 0.020* | 0.015 |
| | (1.89) | (1.44) |
| Market | 0.014 | 0.040*** |
| | (0.98) | (2.94) |
| Gender_Chair (Gender_CEO) | -0.183** | 0.042 |
| | (-2.10) | (0.77) |
| Age_Chair (Age_CEO) | 0.000 | -0.000 |
| | (0.10) | (-0.26) |
| Degree_Chair (Degree_CEO) | -0.003 | 0.024 |
| | (-0.27) | (1.62) |
| Constant | 3.888*** | 4.413*** |
| | (6.08) | (6.93) |
| Ind &Year | Yes | Yes |
| Observations | 2574 | 2857 |
| R-squared | 0.1055 | 0.1065 |

Notes: Robust t-statistic is reported in parentheses.

***, ** and * indicate the significance at the level of 1%, 5% and 10%, respectively.

## 6.1. Heterogeneity of market competition

When firms try to survive and develop in their market, the primary issue they face is market competition. When market competition becomes fierce, on the one hand, enhancing employee welfare, professional training and other forms of employee social responsibility will consume corporate resources and increase enterprise costs, weakening a corporate competitive position [9]. On the other hand, this continuously increasing competitive pressure triggers more constraints on corporate market share and pricing strategy. As a result, firms' potential profits are capped dramatically, and they might also suffer a survival crisis. Therefore, when faced with drastic market competition, senior executives have far greater motivation to rescue firms from

collapse than to fulfill employee social responsibility due to their home bias. Relevant profits are lower than costs in this case. Therefore, the promotion effect of senior executives' home bias on the fulfillment of employee social responsibility will be weakened by adversely fierce market competition. Specifically, in industries with intensive market competition, there is no positive relationship between the home bias of senior executives and the corporate fulfillment of employee social responsibility. Such a positive relationship exists only in less competitive industries.

Hence, we adopt the broadly used Herfindahl-Hirschman Index (HHI) to measure the intensity of product market competition [10]. The specific computation is as follows:

$$HHI = \sum_{i=1}^{N} (Sale_i / SALE)^2 \tag{4}$$

In this equation, i represents each firm within an industry. $Sale_i$ is the sale amount of each firm within the industry, while SALE is the total sale amount of the whole industry. N is the total number of firms in the industry. An industry is more competitive if there are more firms in this industry and each firm has a small market share. Therefore, a smaller HHI means a higher degree of market competition for an industry. Based on the industrial annual median of the HHI, the samples are divided into groups with intensive market competition and groups with less market competition and are regressed.

The regression results are displayed in Table 11. For the samples with lower market competition, Nati_Chair and Nati_CEO are both positive and significant at the 5% and 10% levels, respectively. For samples with intensive market competition, Nati_Chair and Nati_CEO are no longer significant. These results show that the degree of market competition affects corporate performance. Only when a market is less competitive will corporate profits and operating cash flows be more stable [56]. In addition, in a less competitive market, enterprises can exist in better circumstances, be less sensitive to costs and be more willing to fulfill employee social responsibility. Firms balance costs and profits based on market competition and choose to fulfill employee social responsibility only when it is easier for them to survive. Therefore, the promoting effect of senior executives" home bias on fulfilling employees' social responsibility is more significant when a market is less competitive.

## 6.2. Heterogeneity of government subsidies

One of the corporate costs of fulfilling employee social responsibility is a higher level of staff redundancy (Table 10). For firms with sufficient government subsidies, it is a mutual benefit with the government to fulfill employee social responsibility and to take on government employment tasks in senior executives' hometown. If firms fulfill corporate employee social responsibility completely based on their hometown identity, the promotion effect of senior executives' home bias will not be affected by whether the government subsidizes costs or by the amount of government subsidies. In contrast, if there is benefit exchange between firms and the government, the costs and profits related to such a corporation will be taken into account. When government subsidies are not sufficient, corporate senior executives tend to abandon employee social responsibility after this trade-off, even if they have a hometown identity.

To prove the two assumptions above, we divide our samples into groups with a high government subsidy level and a low government subsidy level based on the industry-year median of government subsidies that firms receive. We then regress these groups. Table 12 displays the corresponding results. Clearly, with regard to groups with a high government subsidy level, Nati_Chair and Nati_CEO are both positive and significant at the 5% level, but the coefficients are no longer significant for groups with a low government subsidy level. These results show that when there is a higher level of governmental subsidies, senior executives' home bias more

**Table 11. Market competition, senior executive home bias and employee social responsibility.**

| | Chairperson | | CEO | |
|---|---|---|---|---|
| | **Intensive market competition** | **Lower market competition** | **Intensive market competition** | **Lower market competition** |
| Nati_Chair | -0.015 | 0.051** | | |
| | (-0.80) | (2.53) | | |
| Nati_CEO | | | -0.004 | 0.060*** |
| | | | (-0.20) | (3.45) |
| Size | 0.013 | 0.053*** | 0.029*** | 0.053*** |
| | (1.56) | (5.42) | (3.59) | (6.48) |
| Lev | 0.032 | -0.118* | 0.009 | -0.093 |
| | (0.48) | (-1.73) | (0.15) | (-1.61) |
| Roa | 0.064 | 0.107 | 0.030 | -0.034 |
| | (0.30) | (0.46) | (0.14) | (-0.21) |
| Fcf | 0.037 | -0.063 | -0.018 | 0.017 |
| | (0.29) | (-0.50) | (-0.14) | (0.14) |
| Top1 | -0.183** | -0.093 | -0.064 | 0.031 |
| | (-2.09) | (-1.07) | (-0.76) | (0.47) |
| Stru | 0.051 | 0.105 | -0.083 | 0.020 |
| | (0.60) | (1.31) | (-1.05) | (0.30) |
| PerGDP | 0.011 | -0.141*** | 0.042 | -0.051 |
| | (0.27) | (-3.56) | (1.09) | (-1.48) |
| Law | -0.010* | 0.019*** | -0.007 | 0.015*** |
| | (-1.75) | (3.22) | (-1.06) | (2.87) |
| Market | 0.011 | 0.010 | 0.003 | 0.003 |
| | (1.00) | (0.88) | (0.27) | (0.25) |
| Gender_Chair (Gender_CEO) | -0.038 | 0.041 | -0.021 | 0.068* |
| | (-1.37) | (0.96) | (-0.67) | (1.65) |
| Age_Chair (Age_CEO) | 0.002* | 0.000 | 0.002 | -0.001 |
| | (1.95) | (0.21) | (1.18) | (-0.87) |
| Degree_Chair (Degree_CEO) | -0.004 | 0.001 | -0.018* | 0.017 |
| | (-0.44) | (0.07) | (-1.76) | (1.58) |
| Constant | 1.034** | 1.390*** | 0.555 | 0.595* |
| | (2.57) | (3.63) | (1.56) | (1.68) |
| Ind&Year | Yes | Yes | Yes | Yes |
| Observations | 1302 | 1374 | 1340 | 1637 |
| R-squared | 0.2197 | 0.2643 | 0.1872 | 0.2367 |

Notes: Robust t-statistic is reported in parentheses.

***, ** and * indicate the significance at the level of 1%, 5% and 10%, respectively.

clearly promotes corporate social responsibility. Governmental subsidies are an important method of government intervention [57], and governments tend to use subsidies to exchange benefits with enterprises [58]. The results in Table 12 confirm that there is benefit exchange between enterprises and governments. Senior executives choose to fulfill employee social responsibility and take on employment tasks in exchange for more government subsidies.

## 6.3. Heterogeneity of the nature of enterprise property rights

There are distinct differences in the operating goals and government relationships of state-owned enterprises and nonstate-owned enterprises in China. In state-owned enterprises,

**Table 12. Government subsidies, senior executive home bias and employee social responsibility.**

| | Chairperson | | CEO | |
|---|---|---|---|---|
| | **High government subsidy** | **Low government subsidy** | **High government subsidy** | **Low government subsidy** |
| Nati_Chair | 0.038** | 0.003 | | |
| | (2.22) | (0.10) | | |
| Nati_CEO | | | 0.037** | 0.031 |
| | | | (2.49) | (1.32) |
| Size | 0.028*** | 0.026** | 0.039*** | 0.035*** |
| | (3.50) | (2.01) | (5.53) | (2.66) |
| Lev | -0.020 | -0.123 | -0.040 | -0.085 |
| | (-0.32) | (-1.54) | (-0.76) | (-1.04) |
| Roa | 0.198 | -0.081 | 0.007 | -0.015 |
| | (1.01) | (-0.28) | (0.05) | (-0.05) |
| Fcf | 0.043 | -0.164 | 0.083 | -0.192 |
| | (0.37) | (-1.07) | (0.76) | (-1.28) |
| Top1 | -0.125 | -0.053 | 0.037 | -0.066 |
| | (-1.62) | (-0.49) | (0.57) | (-0.68) |
| Stru | 0.047 | 0.159 | -0.097 | 0.170 |
| | (0.65) | (1.42) | (-1.59) | (1.60) |
| PerGDP | -0.064* | -0.063 | -0.017 | -0.018 |
| | (-1.86) | (-1.26) | (-0.54) | (-0.36) |
| Law | 0.008 | 0.001 | 0.006 | 0.006 |
| | (1.45) | (0.14) | (1.16) | (0.81) |
| Market | 0.002 | 0.021 | 0.004 | 0.006 |
| | (0.26) | (1.54) | (0.45) | (0.43) |
| Gender_Chair (Gender_CEO) | 0.019 | -0.023 | 0.044 | 0.003 |
| | (0.55) | (-0.61) | (1.31) | (0.07) |
| Age_Chair (Age_CEO) | 0.001 | 0.002 | -0.000 | 0.001 |
| | (1.20) | (1.24) | (-0.20) | (0.88) |
| Degree_Chair (Degree_CEO) | -0.000 | -0.004 | 0.009 | -0.023* |
| | (-0.00) | (-0.29) | (0.97) | (-1.73) |
| Constant | 1.158*** | 1.094* | 0.719** | 0.683 |
| | (3.39) | (1.91) | (2.23) | (1.41) |
| Ind&Year | Yes | Yes | Yes | Yes |
| Observations | 1808 | 868 | 2068 | 909 |
| R-squared | 0.2627 | 0.2472 | 0.2235 | 0.2055 |

Notes: Robust t-statistic is reported in parentheses.

***, ** and * indicate the significance at the level of 1%, 5% and 10%, respectively.

senior executives are not only professional managers but also government officers. For most senior executives, their identity as a government officer entails a stronger bias [59]. Senior executives who work in the hometown of their state-owned enterprise tend to have a stronger sense of hometown identity because of their government officer status. First, according to Zhang et al. [53], senior executives of state-owned enterprises usually play a role in local political and strategic decisions and are able to act due to their identification. Second, this type of senior executive is confronted by a stricter social comment constraint because such an identity exposes executives to the attention and supervision of their hometown friends and relatives as well as the Communist Party. This social constraint increases the reputation costs incurred

when charged with not treating employees well and ignoring employee social responsibility. The damage costs of social networks increase as well. Finally, the natural relationship between state-owned enterprises and governments makes these firms balance economic profits and performance costs differently than nonstate-owned enterprises, since the pursuit of economic profits is not the primary goal of state-owned enterprises.

Based on the nature of corporate property rights, we divide our sample into state-owned enterprises and nonstate-owned enterprises and then regress them. Table 13 displays these regression results. For the state-owned enterprises, Nati_Chair and Nati_CEO are both positive and significant at the 10% and 1% levels, respectively. For the nonstate-owned enterprises,

**Table 13. Nature of property rights, senior executive home bias and employee social responsibility.**

| | Chairperson | | CEO | |
|---|---|---|---|---|
| | State-owned enterprises | Nonstate-owned enterprises | State-owned enterprises | Nonstate-owned enterprises |
| Nati_Chair | 0.053* | 0.012 | | |
| | (1.91) | (0.72) | | |
| Nati_CEO | | | 0.075*** | 0.024 |
| | | | (3.02) | (1.25) |
| Size | 0.041*** | 0.029*** | 0.042*** | 0.034*** |
| | (3.26) | (3.37) | (3.26) | (3.58) |
| Lev | -0.056 | -0.017 | -0.111 | -0.059 |
| | (-0.66) | (-0.28) | (-1.40) | (-0.92) |
| Roa | -0.143 | 0.158 | -0.192 | 0.060 |
| | (-0.49) | (0.84) | (-0.72) | (0.32) |
| Fcf | -0.246 | -0.027 | 0.022 | -0.089 |
| | (-1.42) | (-0.25) | (0.13) | (-0.78) |
| Top1 | 0.024 | -0.143* | -0.017 | -0.127 |
| | (0.21) | (-1.86) | (-0.15) | (-1.51) |
| Stru | -0.016 | 0.093 | -0.086 | 0.056 |
| | (-0.13) | (1.25) | (-0.74) | (0.66) |
| PerGDP | -0.005 | -0.063 | 0.011 | -0.062 |
| | (-0.12) | (-1.63) | (0.27) | (-1.43) |
| Law | 0.000 | 0.004 | 0.011 | 0.006 |
| | (0.04) | (0.77) | (1.53) | (0.97) |
| Market | 0.008 | 0.005 | -0.004 | 0.003 |
| | (0.60) | (0.46) | (-0.28) | (0.27) |
| Gender_Chair (Gender_CEO) | -0.100** | 0.051* | -0.015 | 0.005 |
| | (-2.07) | (1.65) | (-0.17) | (0.12) |
| Age_Chair (Age_CEO) | -0.002 | 0.002** | -0.003 | 0.003*** |
| | (-1.14) | (2.55) | (-1.40) | (2.85) |
| Degree_Chair (Degree_CEO) | -0.023 | -0.001 | -0.013 | 0.000 |
| | (-1.34) | (-0.12) | (-0.82) | (0.02) |
| Constant | 0.485 | 1.067*** | 0.249 | 1.232*** |
| | (0.93) | (2.62) | (0.53) | (2.62) |
| Ind&Year | Yes | Yes | Yes | Yes |
| Observations | 965 | 1534 | 978 | 1386 |
| R-squared | 0.1992 | 0.2967 | 0.1915 | 0.2703 |

Notes: Robust t-statistic is reported in parentheses.

***, ** and * indicate the significance at the level of 1%, 5% and 10%, respectively.

however, Nati_Chair and Nati_CEO are no longer significant. Therefore, the promotion effect of home bias on the corporate fulfillment of employee social responsibility is stronger for state-owned enterprises.

In China, state-owned enterprises are controlled by the government. The evaluation of senior executives in state-owned enterprises focuses more on whether they undertake various governmental tasks than their economic effectiveness [60]. It is taken for granted that state-owned enterprises should play a critical role in improving employment [53, 61]. As a result, in state-owned enterprises, their home bias gives senior executives a stronger inclination and ability to undertake more employee social responsibility, resulting in the improved fulfillment thereof.

The results of our heterogeneity analysis further confirm that senior executives' home bias promotes the fulfillment of employee social responsibility due to benefit exchange. When corporate economic profits are not optimistic, enterprises consider the fulfillment of employee social responsibility more costly and thus cease performing it. Only when there are more benefits, i.e., rewards, are local senior executives motivated and willing to better fulfill employee social responsibility.

## 7. Conclusion and suggestions

Currently, it is an important and highly advocated issue that firms should actively fulfill social responsibility to promote their own sustainable development and further promote society's complete sustainable development. Within corporate social responsibility, employee social responsibility is an essential component. Therefore, both theoretical scholars and practical businesses focus closely on how to enhance the corporate fulfillment of employee social responsibility to reduce labor disputes and, hence, to keep society stable. In this paper, we systematically illustrate the impacts of the home bias of senior executives on corporate employee social responsibility using our sample of all A-share listed companies during 2006–2019. We find that both the home bias of chairpersons and that of CEOs can significantly encourage the corporate fulfillment of employee social responsibility. Moreover, this conclusion is robust, based on a series of robustness tests. Through further analysis, we illustrate how senior executives' home bias can encourage firms to perform employee social responsibility, significantly reduce their corporate staff turnover rate, stabilize employment and attract additional government subsidies. However, along with their profits, the fulfillment of employee social responsibility also raises firms' costs, such as an increased level of staff redundancy, when firms take on more government employment tasks. Therefore, senior executives' home bias promotes the fulfillment of employee social responsibility not only due to hometown identification but also via benefit exchange between firms and governments and between firms and employees. This inference is verified by our heterogeneity analysis, where we find that the promotion effect of home bias disappears when a market is seriously competitive, when a government does not subsidize enough or in nonstate-owned enterprises. This finding can be explained as follows: When corporate economic profits become influenced, firms become more concerned with the costs incurred by performing employee social responsibility and are thus likely to abandon such responsibility.

This paper has a certain theoretical value. First, we incorporate home bias, which reflects personal features, into our research framework of the potential motivations for employee social responsibility to prove the promoting effect of senior executives' home bias on the corporate fulfillment of employee social responsibility, thereby filling a research gap in the literature on employee social responsibility. Second, based on previous research, we incorporate identification theory into our theoretical framework of senior executives' home bias to analyze

how home bias functions from the perspective of identification. However, we do not believe that home bias influences corporate behavior merely because of identification. Therefore, we also take into account the interest games between enterprises and their hometown government, society or the public; we find that benefit exchange exists. Accordingly, this work not only extends the theoretical boundary of research on home bias but also enriches the literature on home bias.

This paper also has important practical significance. Currently, the Chinese government is strongly advocating enterprises to actively fulfill employee social responsibility to build harmonious labor relations. As this paper proves that senior executives' home bias promotes the corporate fulfillment of employee social responsibility via the distinct approaches of identification and benefit exchange, our results have specific implications for governments searching for ways to drive enterprises to actively fulfill employee social responsibility, and improve currently disharmonious labor relations. In addition, this paper can be referred to when making relevant policies. The COVID-19 pandemic has shocked many labor markets worldwide and triggered many more labor disputes. Recently, the International Labor Organization (ILO) has advised countries to actively take measures to protect employees and support enterprises. Thus, our findings on employee social responsibility in the Chinese market also have beneficial implications for other emerging markets.

Based on the conclusions we have obtained, we make the following feasible suggestions: First, to guide and encourage enterprises to fulfill employee social responsibility, governments should steadily improve laws and regulations, set corresponding rules for awards and punishments and strengthen supervision. We find that the promoting effect of home bias is greater because of benefit exchange between enterprises and local governments and that this promoting effect becomes more obvious when enterprises receive more governmental subsidies. This positive relationship between home bias's promoting effect and governmental subsidies shows that only when enterprises gain enough subsidies, i.e., enough benefits, will enterprises take greater employee social responsibility and absorb more employment at the cost of abundant staff. Consequently, governments should take more encouraging measures, such as granting more subsidies or tax preferences to firms that fulfill employee social responsibility more actively. At the same time, governments can award such enterprises properly, developing a positive guiding effect for the public. On the other hand, we find that in private enterprises and those in a rather fiercely competitive market, senior executives' home bias does not have a promoting effect on the fulfillment of employee social responsibility. This conclusion shows that when corporate economic profits are threatened, enterprises will prioritize the related costs and cease fulfilling employee social responsibility. Therefore, to lead enterprises to increase investments in employee social responsibility, the government should also set rules for punishments (such as fines, etc.) and punish enterprises that do not fulfill employee social responsibility as needed. In addition, to strengthening supervision and encouraging enterprises to undertake more employee social responsibility, governments should authorize labor unions or other intermediaries to supervise a wider range of objects.

Second, enterprises should be guided to improve their knowledge and awareness in regard to fulfilling employee social responsibility. As the promoting effect of senior executives' home bias is primarily the result of benefit exchange between enterprises and employees or governments, governments should guide enterprises properly and make them understand their employees' importance in corporate sustainable development as well as the necessity of fulfilling employee social responsibility. The government should make it clear to enterprises that benefit exchange is not the most important advantage of fulfilling employee social responsibility. Meanwhile, the government should also offer frequent educational training to enterprise senior executives to improve their awareness of employee social responsibility.

The main limitations of this study and some future research directions are as follows: First, our work might be too narrow and insufficiently comprehensive to show that the origin place of senior executives is consistent enough with the firm-registered place to be used as the indicator for home bias. In future research, to improve measurement rationality and accuracy, a more comprehensive indicator should be built, probably by including the length of senior executives' residency in their origin place, the religious culture thereof and other aspects. Second, there might be other potential channels through which senior executives' home bias influences corporate behavior, in addition to the interest exchange view and the theory of identity we utilize in this paper. Future studies can thus explore these possible channels and further broaden the theoretical framework for research on the effects of senior executives' home bias.

## Supporting information

**S1 Data.**
(ZIP)

**S1 Graphical abstract. "+" (red) is indicated that the impact of senior executives' home bias on employee social responsibility (ESR) is significantly positive.** Both identification (left circle) and benefit exchange (right circle) are the motivation of this relationship, and benefit exchange is the more important motivation. The cost-benefit analysis and heterogeneity analysis further verify the benefit exchange motivation.
(TIF)

## Author Contributions

**Data curation:** Wenjing Sun.

**Formal analysis:** Tingting Zhang.

**Project administration:** Tingting Zhang, Xing Rong.

**Writing – original draft:** Tingting Zhang, Wenjing Sun.

**Writing – review & editing:** Xing Rong, Jingyi Mu.

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
