## [Decision Letter · Decision Letter 0]

31 Aug 2022

PONE-D-22-15466Home Bias and Employee Social Responsibility: Identification v.s. Benefit ExchangePLOS ONE

Dear Dr. Rong,

Thank you for submitting your manuscript to PLOS ONE. After careful consideration, we feel that it has merit but does not fully meet PLOS ONE’s publication criteria as it currently stands. Therefore, we invite you to submit a revised version of the manuscript that addresses the points raised during the review process.

Please submit your revised manuscript by Oct 15 2022 11:59PM.  If you will need more time than this to complete your revisions, please reply to this message or contact the journal office at plosone@plos.org. Please include the following items when submitting your revised manuscript:A rebuttal letter that responds to each point raised by the academic editor and reviewer(s). You should upload this letter as a separate file labeled 'Response to Reviewers'.A marked-up copy of your manuscript that highlights changes made to the original version. You should upload this as a separate file labeled 'Revised Manuscript with Track Changes'.An unmarked version of your revised paper without tracked changes. You should upload this as a separate file labeled 'Manuscript'.

We look forward to receiving your revised manuscript.

Kind regards,

Tai Ming Wut

Academic Editor

PLOS ONE

Journal Requirements:

“YES

This work was partially supported by the National Science Foundation for Young Scientists of China [grant No. 71902163], the Humanities and Social Sciences Youth Foundation of Ministry of Education of China [grant No. 19YJC630218] and Fundamental Research Funds for the Central Universities [grant No. JBK2202015].”

Reviewers' comments:

Reviewer's Responses to Questions

**Comments to the Author**

1. Is the manuscript technically sound, and do the data support the conclusions?

Reviewer #1: Yes

Reviewer #2: Yes

2. Has the statistical analysis been performed appropriately and rigorously? 

Reviewer #1: Yes

Reviewer #2: Yes

3. Have the authors made all data underlying the findings in their manuscript fully available?

Reviewer #1: Yes

Reviewer #2: Yes

4. Is the manuscript presented in an intelligible fashion and written in standard English?

Reviewer #1: Yes

Reviewer #2: Yes

5. Review Comments to the Author

Reviewer #1: I enjoyed reading the article "Home Bias and Employee Social Responsibility: Identification v.s. Benefit Exchange". This well-written article discusses an intriguing topic. It is imperative that this study be conducted. However, some issues need to be addressed.

Abstracts should include a concise description of the study's purpose and implications. It is difficult to follow the abstract because it is very long and complicated. The abstract should be revised and improved by the authors. To better understand their study at a glance, authors are also recommended to provide a graphical abstract.

The introduction section should include a clear explanation of the study's context and research objective. The research gap must also be narrowed after analyzing the previous studies. The claim that the study contributed to the field is not adequately supported.

There are some improvements that need to be made to the literature review. The authors missed the latest literature as no study from 2022 is cited, and only one study from 2021 is included in the review. How valid are the claims made by authors based on old literature? To correctly identify the research gap, the authors should rewrite the literature.

Although the results of this study are relevant, they have not been adequately discussed, and they have not been supported by significant and recent literature. The most recent research articles should also be used to support your findings.

Lastly, the authors should improve the quality of their conclusion. Although the authors have provided some recommendations, they should concentrate on the conclusions supported by their findings. The authors should provide future directions for research as well as practical implications.

English writing and grammar have some serious problems. Professional proofreaders are recommended for the authors.

Reviewer #2: I am pleased to review the manuscript. The authors have done well in structuring their study. Overall, the sections are elaborately written. Hence, I believe that this study can be considered for publication provided the authors are ready to revise their manuscript as per the major comments provided below:

• Abstract: The abstract should include at the beginning to provide a brief background of the study.

• The keywords appear to be more like long phrases. I suggest revising the list of keywords and making them more concise.

• Introduction: The contributions of the study should be precisely highlighted.

• Please provide a sentence or two with more details regarding the contributions of this work and main outcomes from the study.

• Please add study measures in the revised manuscript.

• Conclusion: Please mention the limitations of the study and mention the future research direction as well.

• Proofread the manuscript to correct grammatical errors and smoothen the flow of contents.

Note> It systematizes and shortens many of the paragraphs because the great length of the paper makes the essence lost.

Look forward to receiving the revised version soon.

6. PLOS authors have the option to publish the peer review history of their article (what does this mean?). If published, this will include your full peer review and any attached files.

Reviewer #1: **Yes: **Dr. Shahid Ali

Reviewer #2: No

---

## [Author Response · Author response to Decision Letter 0]

20 Oct 2022

Responses to Reviewer1:

Home Bias and Employee Social Responsibility: Identification v.s. Benefit Exchange

Dear Reviewer:

Our sincere thank you for the very constructive comments on our manuscript. Your constructive comments on the paper were extremely valuable in improving the quality of the paper. We have carefully revised the manuscript and adopted all of your suggestions. In what follows, your comments are shown in italics and followed by our point-by-point responses. All the page numbers refer to those in the revision. Changes made according to the Comments in the revised manuscript are highlighted in blue color of the revised paper for ease of your reference. We believe that you will find this report and our revised manuscript have satisfactorily addressed the issues you have raised, and we thank you in advance for your valuable time in reviewing the manuscript.

Reviewer’s Comment 1:

Abstracts should include a concise description of the study's purpose and implications. It is difficult to follow the abstract because it is very long and complicated. The abstract should be revised and improved by the authors. To better understand their study at a glance, authors are also recommended to provide a graphical abstract.

Authors’ Response (Comment 1):

Thank you for the reviewer's valuable comments. Based on the suggestion, we have completely rewritten the abstracts. In the revised summary, we also briefly illustrate the purpose and implication of this research to make it easier for readers to understand the abstract. Meanwhile, we add a graphical abstract according to the reviewer’s suggestion. The abstract has been revised as follows, and is marked in blue in the revised manuscript (Line27-59 on p. 2-3)：

Abstract: Employees, as the most valuable assets and critical sources of competitive advantage in enterprises, are among the important stakeholders in enterprises. Employee social responsibility (ESR) has been a continually important research interest in the field of enterprise social responsibility. However, in the literature, few studies explore how personal features affect employee social responsibility. Thus, sampling China’s listed companies from 2006 to 2019, we investigate how the home bias of senior executives influences enterprises’ employee social responsibility. We identify home bias based on whether a chairperson’s or CEO’s hometown matches the firm’s registration place. Three main results are obtained. First, the home bias of both CEOs and chairpersons can improve the corporate fulfillment of employee social responsibility. Second, further cost‒benefit analysis shows that this result is due to not only identification but also benefit exchange. Although senior executives’ home bias significantly decreases employee turnover rate, enterprises absorb more employment, which significantly increases their redundant personnel costs. Therefore, firms should balance the potential benefits and costs incurred by home bias via trade-off. Third, in firms facing less market competition, firms with more governmental subsidies or state-owned firms, senior executives’ home bias has a more significant promoting effect on the fulfillment of ESR, supporting the view of benefit exchange. Accordingly, by extending theories on the effects of senior executives’ home bias and enriching the ESR literature, this paper has important practical value, our findings can guide and promote firms to perform ESR while actively complying with a national policy for stabilizing employment and ensuring people’s well-being.

Graphical abstract【The figure shows in Fig.1 in the manuscript】

Reviewer’s Comment 2:

The introduction section should include a clear explanation of the study's context and research objective. The research gap must also be narrowed after analyzing the previous studies. The claim that the study contributed to the field is not adequately supported.

Authors’ Response (Comment 2):

We appreciate for your comment on raising this very important issue. We realize that the previous introduction did not clearly show our research background or purpose. Since we did not review existing related literature properly, this study’s contribution to the field is not emphasized enough either. Neither do these shortfalls improve the research quality, nor do them help readers understand our conclusions. Therefore, based on the reviewer's suggestion, we have reorganized and rewritten the introduction. In the new introduction, we describe the research background clearly, summarize existing literature in detail and point out the research gap among current literature. We then come up with our research purpose based on analysis above. The clear explanation of the study's context and research objective has been revised as follows, and is marked in blue in the revised manuscript (Line 63-144 on p. 3-7) :

Employees are the main stakeholders of enterprise social responsibility. The protection of employee rights and interests is the most direct and important part of enterprise social responsibility[1] and has gained extensive attention from international organizations, such as the International Labor Organization and the International Standardization Organization, which aim to standardize enterprise social responsibility[2]. Employee social responsibility is mainly reflected in the care and concerns of firms for their employees. For instance, firms can perform social responsibility by providing employees with competitive salaries while ensuring their welfare and safe working conditions[3] or by offering employees sufficient training and humane treatment[4]. By treating employees well and securing their rights and interests, firms can not only promote more employee organizational citizenship behavior[2] and innovation[5] but also achieve higher total factor productivity(TFP)[6], improve future performance[6] and obtain higher growth potential[7].

However, compared to donations, product services and other social responsibilities that the media and public are familiar with, employee social responsibility is more internal and tends to be ignored by firms. The basic performance of employee social responsibility in many firms is rather unsatisfactory. International Labor Organization (ILO) data show that since 2010, there have been at least 44000 work stoppages worldwide; however, more seriously, these data may not cover all economic activities and all geographic areas. The number of disputes in Chinese companies has steadily risen, while in many countries and regions of the world, labor-related conflicts and mobilizations are becoming ever more intense and frequent [8]. In China, according to statistics of the Ministry of Human Resources and Social Security, by the end of 2021, there had been approximately 2.631 million labor disputes, and this number had increased by 140.27% on a year-on-year basis. In these disputes, most problems concern poor working conditions, employee health and safety, limited development potential and unsatisfactory welfare. It is an unarguable fact that disputes arising from labor-management conflicts remain at a high level. At the same time, data also reveal that there are still a large number of companies globally that seriously lack any awareness of consciously fulfilling employee social responsibility. The current situation thus not only restricts the sustainable development of enterprises but also poses a great challenge to securing people's livelihood and maintaining social harmony and stability. Therefore, in-depth research on the fulfillment of employee social responsibility has become an important topic that currently has both theoretical significance and practical value.

In recent years, academics have attempted to distinguish employee social responsibility and general social responsibility for specialized research. The literature has confirmed that market competition[9,10], regional Confucian cultural intensity[11], corporate hypocrisy[12], the establishment of labor-management committees[13], population mobility[14], and family business restructuring[3] are important factors influencing the fulfillment of employee social responsibility. Unfortunately, most of these studies have been based on external factors, while explorations at the individual level have been less common. Higher-order theory emphasizes that a team of senior executives plays a central role in an organization and that the personal features of these team members have a significant impact on corporate decisions [15]. Therefore, an important question arises: when senior executives are making decisions on employee social responsibility, does their personal bias affect responsibility fulfillment?

Home bias represents an individual’s inherent favoritism toward his or her hometown and has been widely discussed worldwide. Governmental officers tend to distribute more political resources to their hometown [16]. Fund managers generally hold stocks of firms from their hometown in a greater proportion than average[17]. Senior executives are inclined to establish more subsidiary firms [18] and make more acquisitions and investments[19,20,21] in their hometown. However, the literature remains mostly based on the identification economics proposed by Akerlof and Franton (2000; 2002)[22,23]. Identification theory emphasizes individual morality and group responsibility but ignores the social exchanges and interest games driven by profit-seeking instincts among individuals and groups. Because the fulfillment of employee social responsibility cannot benefit firms instantaneously, it is often ignored by the media as well as the public, it is valuable to determine the possible motivations for such performance and to explore how firms engage in trade-off during their interest games with local government and other stakeholders.

Based on the above analysis, we investigate the influences and inner motivations of senior executives regarding the corporate fulfillment of employee social responsibility from the perspective of home bias. We attempt to resolve the following questions: First, how does the home bias of senior executives influence the corporate fulfillment of employee social responsibility? Second, what are the inner motivations driving senior executives’ influence on the corporate fulfillment of employee social responsibility? Is there a benefit-exchange motivation in addition to the identification motivation? To answer these questions, we take all A-share listed companies in China from 2006 to 2019 as our sample and measure senior executives’ home bias based on whether a chairperson’s or CEO’s hometown is consistent with the firm’s registration place. We use the score index for employee social responsibility from the CNRDS dataset to measure corresponding corporate fulfillment, and we systematically investigate the effects of senior executives’ home bias on corporate employee social responsibility. 

Reviewer’s Comment 3:

There are some improvements that need to be made to the literature review. The authors missed the latest literature as no study from 2022 is cited, and only one study from 2021 is included in the review. How valid are the claims made by authors based on old literature? To correctly identify the research gap, the authors should rewrite the literature.

Authors’ Response (Comment 3):

We greatly appreciate for the suggestions. As the reviewer said, we missed the latest literature as no study from 2022 is cited, and only one study from 2021 is included in the review. Such a lack of recency makes our conclusions doubtful. Therefore, we have rewritten the literature review and added loads of the latest researches, especially those from 2020 to 2022. In this way, the research gap should be identified correctly, and conclusions are more convincing. Modified literature reviews are as follows, and is marked in blue in the revised manuscript：

Line 63-76 on p. 3-4: Employees are the main stakeholders of enterprise social responsibility. The protection of employee rights and interests is the most direct and important part of enterprise social responsibility[1] and has gained extensive attention from international organizations, such as the International Labor Organization and the International Standardization Organization, which aim to standardize enterprise social responsibility[2]. Employee social responsibility is mainly reflected in the care and concerns of firms for their employees. For instance, firms can perform social responsibility by providing employees with competitive salaries while ensuring their welfare and safe working conditions[3] or by offering employees sufficient training and humane treatment[4]. By treating employees well and securing their rights and interests, firms can not only promote more employee organizational citizenship behavior[2] and innovation[5] but also achieve higher total factor productivity(TFP)[6], improve future performance[6] and obtain higher growth potential[7].

Line 99-125 on p. 5-6: In recent years, academics have attempted to distinguish employee social responsibility and general social responsibility for specialized research. The literature has confirmed that market competition[9,10], regional Confucian cultural intensity[11], corporate hypocrisy[12], the establishment of labor-management committees[13], population mobility[14], and family business restructuring[3] are important factors influencing the fulfillment of employee social responsibility. Unfortunately, most of these studies have been based on external factors, while explorations at the individual level have been less common. Higher-order theory emphasizes that a team of senior executives plays a central role in an organization and that the personal features of these team members have a significant impact on corporate decisions [15]. Therefore, an important question arises: when senior executives are making decisions on employee social responsibility, does their personal bias affect responsibility fulfillment?

Home bias represents an individual’s inherent favoritism toward his or her hometown and has been widely discussed worldwide. Governmental officers tend to distribute more political resources to their hometown [16]. Fund managers generally hold stocks of firms from their hometown in a greater proportion than average[17]. Senior executives are inclined to establish more subsidiary firms [18] and make more acquisitions and investments[19,20,21] in their hometown. However, the literature remains mostly based on the identification economics proposed by Akerlof and Franton (2000; 2002)[22,23]. Identification theory emphasizes individual morality and group responsibility but ignores the social exchanges and interest games driven by profit-seeking instincts among individuals and groups. Because the fulfillment of employee social responsibility cannot benefit firms instantaneously, it is often ignored by the media as well as the public, it is valuable to determine the possible motivations for such performance and to explore how firms engage in trade-off during their interest games with local government and other stakeholders.

Reviewer’s Comment 4:

Although the results of this study are relevant, they have not been adequately discussed, and they have not been supported by significant and recent literature. The most recent research articles should also be used to support your findings.

Authors’ Response (Comment 4):

Thank you for the reviewer's valuable comments. As the reviewer said, previous empirical results did lack adequate discussions and we did not cite important or latest literature to support our conclusions, making our findings kind of untrustworthy and incomprehensible. To improve conclusions’ quality, we have discussed about each main empirical result more thoroughly. Meanwhile, we have added important recent studies to support our analysis, to make empirical results more convincing. Modified contents are marked in blue in the revised manuscript：

Line 435-453 on page 23-24, we add related supportive literature for Table 3 and supplement discussions about our results. Details are as follows:

Zhu et al. (2022)[36] find that local senior executives influence enterprises to fulfill social responsibility. Table 3 further proves Zhu’s finding by showing our results regarding employee social responsibility, a subsidiary social responsibility. Although our findings show that the influence of home bias uses the channel of identification[20.37], we do not think home bias functions only because of emotional identification (senior executives want to reward their hometown)[38] or hometown social network constraints [18]. Hejjas et al. (2019)[39] show that a complex combination of public needs, governmental expectations and other reasons encourages an increasing number of enterprises to fulfill social responsibility. To some extent, Hejjas et al.’s finding also reflects the complexity of enterprises’ motivations to fulfill social responsibility, an important one of which is gaining benefits and resources [40]. We suggest that the identity of fellow hometown residents gives a senior executive a wider and more convenient social network and enables him or her to play an interest game with employees or the relevant government to obtain corresponding benefits based on the premise of fulfilling social responsibility. Therefore, the conclusions we have obtained might be due to benefit exchange, which has never been discussed in the literature. Via further analysis, we thus deeply discuss how senior executives’ home bias promotes the fulfillment of employee social responsibility due to benefit-exchange motivation.

Because results from Table 4 to Table 8 are about robustness test, relevant literature are not added.

Line 600-609 on page36, we add supportive literature and discussions about Table 9. Details are as follows:

Generally, employees will gain a sense of identity when they perceive the corporate fulfillment of social responsibility[45,46]. Their sense of identity with their enterprise reflects that employees are proud of and loyal to their enterprise[47]. The results in Table 9 further prove that when senior executives’ home bias promotes enterprises to better fulfill employee social responsibility, employees are better able to perceive this kind of improvement themselves, to develop a stronger sense of loyalty and to be less likely to resign. Consequently, enterprises can build a more stable team of employees, which fosters innovative corporate development[48]. This advantage is one of the benefits of promoting the effect of senior executives’ home bias on fulfilling employee social responsibility.

Line 650-661 on page39-40, we add supportive literature and discussions about Table 10. Details are as follows:

Enterprises usually regard the fulfillment of employee social responsibility and other internal responsibilities as cost-incurring behaviors[3]. Our results in Table 10 show that while senior executives’ home bias promotes the fulfillment of employee social responsibility, it also produces a higher level of redundant personnel. Accordingly, such enterprises accept a certain level of political burden[54], undertake loads of employment tasks[54], and incur additional costs.

In summary, benefit-cost analysis reveals that the significant promoting effect of home bias on fulfilling employee social responsibility stems mostly from benefit exchange between senior executives and enterprises, governments or employees. Senior executives’ fulfillment of employee social responsibility can only be encouraged and promoted with vast benefits such as rewards. This condition is clearly shown in the next section’s heterogeneity analysis.

Line 703-711 on page43, we add supportive literature and discussions about Table 11. Details are as follows:

These results show that the degree of market competition affects corporate performance. Only when a market is less competitive will corporate profits and operating cash flows be more stable[56]. In addition, in a less competitive market, enterprises can exist in better circumstances, be less sensitive to costs and be more willing to fulfill employee social responsibility. Firms balance costs and profits based on market competition and choose to fulfill employee social responsibility only when it is easier for them to survive. Therefore, the promoting effect of senior executives’’ home bias on fulfilling employees’ social responsibility is more significant when a market is less competitive.

Line 732-739 on page 45, we add supportive literature and discussions about Table 12. Details are as follows:

These results show that when there is a higher level of governmental subsidies, senior executives’ home bias more clearly promotes corporate social responsibility. Governmental subsidies are an important method of government intervention[57], and governments tend to use subsidies to exchange benefits with enterprises[58]. The results in Table 12 confirm that there is benefit exchange between enterprises and governments. Senior executives choose to fulfill employee social responsibility and take on employment tasks in exchange for more government subsidies.

Line 769-775 on page48, we add supportive literature and discussions about Table 13. Details are as follows:

In China, state-owned enterprises are controlled by the government. The evaluation of senior executives in state-owned enterprises focuses more on whether they undertake various governmental tasks than their economic effectiveness[60]. It is taken for granted that state-owned enterprises should play a critical role in improving employment[53,61]. As a result, in state-owned enterprises, their home bias gives senior executives a stronger inclination and ability to undertake more employee social responsibility, resulting in the improved fulfillment thereof.

Line 778-783 on page 49, we also add supportive literature and discussions. Details are as follows:

The results of our heterogeneity analysis further confirm that senior executives’ home bias promotes the fulfillment of employee social responsibility due to benefit exchange. When corporate economic profits are not optimistic, enterprises consider the fulfillment of employee social responsibility more costly and thus cease performing it. Only when there are more benefits, i.e., rewards, are local senior executives motivated and willing to better fulfill employee social responsibility.

Reviewer’s Comment 5:

Lastly, the authors should improve the quality of their conclusion. Although the authors have provided some recommendations, they should concentrate on the conclusions supported by their findings. The authors should provide future directions for research as well as practical implications.

Authors’ Response (Comment 5):

Thanks for the comment. Combined with the reviewer's suggestion, we have made a lot of additions and improvements according to your suggestions. We take a deeper look at the results of the paper，and highlight the theoretical contributions and practical implications of the article. Based on the conclusions supported by our findings, we propose detailed and helpful suggestions from the view of government. Details are as follows:

The theoretical contributions and practical implications of the article(Line 811-837 on page51-52):

This paper has a certain theoretical value. First, we incorporate home bias, which reflects personal features, into our research framework of the potential motivations for employee social responsibility to prove the promoting effect of senior executives’ home bias on the corporate fulfillment of employee social responsibility, thereby filling a research gap in the literature on employee social responsibility. Second, based on previous research, we incorporate identification theory into our theoretical framework of senior executives’ home bias to analyze how home bias functions from the perspective of identification. However, we do not believe that home bias influences corporate behavior merely because of identification. Therefore, we also take into account the interest games between enterprises and their hometown government, society or the public; we find that benefit exchange exists. Accordingly, this work not only extends the theoretical boundary of research on home bias but also enriches the literature on home bias.

This paper also has important practical significance. Currently, the Chinese government is strongly advocating enterprises to actively fulfill employee social responsibility to build harmonious labor relations. As this paper proves that senior executives’ home bias promotes the corporate fulfillment of employee social responsibility via the distinct approaches of identification and benefit exchange, our results have specific implications for governments searching for ways to drive enterprises to actively fulfill employee social responsibility, and improve currently disharmonious labor relations. In addition, this paper can be referred to when making relevant policies. The COVID-19 pandemic has shocked many labor markets worldwide and triggered many more labor disputes. Recently, the International Labor Organization (ILO) has advised countries to actively take measures to protect employees and support enterprises. Thus, our findings on employee social responsibility in the Chinese market also have beneficial implications for other emerging markets.

Suggestions from the view of government (Line 838-872 on page52-53)

Based on the conclusions we have obtained, we make the following feasible suggestions: First, to guide and encourage enterprises to fulfill employee social responsibility, governments should steadily improve laws and regulations, set corresponding rules for awards and punishments and strengthen supervision. We find that the promoting effect of home bias is greater because of benefit exchange between enterprises and local governments and that this promoting effect becomes more obvious when enterprises receive more governmental subsidies. This positive relationship between home bias’s promoting effect and governmental subsidies shows that only when enterprises gain enough subsidies, i.e., enough benefits, will enterprises take greater employee social responsibility and absorb more employment at the cost of abundant staff. Consequently, governments should take more encouraging measures, such as granting more subsidies or tax preferences to firms that fulfill employee social responsibility more actively. At the same time, governments can award such enterprises properly, developing a positive guiding effect for the public. On the other hand, we find that in private enterprises and those in a rather fiercely competitive market, senior executives’ home bias does not have a promoting effect on the fulfillment of employee social responsibility. This conclusion shows that when corporate economic profits are threatened, enterprises will prioritize the related costs and cease fulfilling employee social responsibility. Therefore, to lead enterprises to increase investments in employee social responsibility, the government should also set rules for punishments (such as fines, etc.) and punish enterprises that do not fulfill employee social responsibility as needed. In addition, to strengthening supervision and encouraging enterprises to undertake more employee social responsibility, governments should authorize labor unions or other intermediaries to supervise a wider range of objects.

Second, enterprises should be guided to improve their knowledge and awareness in regard to fulfilling employee social responsibility. As the promoting effect of senior executives’ home bias is primarily the result of benefit exchange between enterprises and employees or governments, governments should guide enterprises properly and make them understand their employees’ importance in corporate sustainable development as well as the necessity of fulfilling employee social responsibility. The government should make it clear to enterprises that benefit exchange is not the most important advantage of fulfilling employee social responsibility. Meanwhile, the government should also offer frequent educational training to enterprise senior executives to improve their awareness of employee social responsibility.

Last but not the least, in the original manuscript, we have described limitations and future research directions of the study, but the reviewer did not notice these contents, possibly due to our inaccurate descriptions. We are really sorry for this miss and have emphasized limitations and future research orientations in the revised manuscript in a more noticeable way. Modified conclusions are as follows, and is marked in blue in the revised manuscript (Line873-874 on p. 54)：

The main limitations of this study and some future research directions are as follows: First, our work might be too narrow and insufficiently comprehensive to show that the origin place of senior executives is consistent enough with the firm-registered place to be used as the indicator for home bias. In future research, to improve measurement rationality and accuracy, a more comprehensive indicator should be built, probably by including the length of senior executives’ residency in their origin place, the religious culture thereof and other aspects. Second, there might be other potential channels through which senior executives’ home bias influences corporate behavior, in addition to the interest exchange view and the theory of identity we utilize in this paper. Future studies can thus explore these possible channels and further broaden the theoretical framework for research on the effects of senior executives’ home bias.

Reviewer’s Comment 6:

English writing and grammar have some serious problems. Professional proofreaders are recommended for the authors.

Authors’ Response (Comment 6):

We greatly appreciate for the suggestions. We apologize for the poor English writing and grammar of our manuscript. Based on the reviewer's suggestion, we have involved professional proofreaders for English writing and grammar corrections. We really hope that English writing and grammar have been substantially improved. Please review the revised manuscript. 

Responses to Reviewer2:

Home Bias and Employee Social Responsibility: Identification v.s. Benefit Exchange

Dear Reviewer:

Our sincere thank you for the very constructive comments on our manuscript. Your constructive comments on the paper were extremely valuable in improving the quality of the paper. We have carefully revised the manuscript and adopted all of your suggestions. In what follows, your comments are shown in italics and followed by our point-by-point responses. All the page numbers refer to those in the revision. Changes made according to the comments in the revised manuscript are highlighted in blue color of the revised paper for ease of your reference. We believe that you will find this report and our revised manuscript have satisfactorily addressed the issues you have raised, and we thank you in advance for your valuable time in reviewing the manuscript.

Reviewer’s Comment 1:

Abstract: The abstract should include at the beginning to provide a brief background of the study.

Authors’ Response (Comment 1):

We appreciate for the suggestion. As the reviewer said, There is no brief description of our study’s background in previous abstract, making it harder for readers to understand this paper. Therefore, we have added background illustrations in the modified abstract. Details are as follows, and is marked in blue in the revised manuscript (Line27-31 on p. 2)：

Employees, as the most valuable assets and critical sources of competitive advantage in enterprises, are among the important stakeholders in enterprises. Employee social responsibility (ESR) has been a continually important research interest in the field of enterprise social responsibility. However, in the literature, few studies explore how personal features affect employee social responsibility.

Reviewer’s Comment 2:

The keywords appear to be more like long phrases. I suggest revising the list of keywords and making them more concise.

Authors’ Response (Comment 2):

We appreciate for the suggestion. After the reviewer's guidance, we realize that the list of keywords is not concise enough. Consequently, we make some changes and modified details are as follows, and is marked in blue in the revised manuscript (Line 60 on p. 3)：

Key words: senior executives, home bias, employee social responsibility, benefit exchange, identification

Reviewer’s Comment 3:

Introduction: The contributions of the study should be precisely highlighted.

Reviewer’s Comment 4:

Please provide a sentence or two with more details regarding the contributions of this work and main outcomes from the study.

Authors’ Response (Comment 3-4):

Thank you to the reviewer for pointing out the problem of the contributions of this work and main outcomes from the study. Because the third comment and the fourth comment are both about study contributions and conclusions, we respond the two to the reviewer together. Under the reviewer’s guidance, we find that the part of study conclusions and the one of contributions need improvements. For example, conclusions were too long and too complicated, and contributions were not emphasized properly. After further readings and based on the reviewer’s comments, we have trimmed and improved study conclusions and contributions. We have deleted those minor contents and rewritten study contributions more precisely. We believe that conclusions and contributions of this work become clearer and more understandable after modifications. Details are as follows, and is marked in blue in the revised manuscript (Line169-192 on p.8-9 )：

First, the home bias of both CEO and chairperson can improve firms’ fulfillment of employee social responsibility. Second, further cost‒benefit analysis reveals that this result is due to not only identification but also benefit exchange. Although senior executives’ home bias significantly decreases employee turnover rate, firms absorb more employment, significantly increasing their redundant personnel costs. Therefore, firms should balance the potential benefits and costs of home bias via trade-off. Third, in firms facing less market competition, firms with more governmental subsidies or state-owned firms, senior executives’ home bias has a more significant promoting effect on the fulfillment of employee social responsibility, supporting the view of benefit exchange. The innovation and contribution of this paper are thus mainly reflected in the following: First, in the context of China’s special culture, we incorporate home bias, which reflects personal features, into our research framework of the potential motivations for employee social responsibility, filling a research gap in the literature on employee social responsibility. Second, in addition to supporting the identification motivation related to home bias, we creatively propose that senior executives’ home bias promotes the fulfillment of employee social responsibility due to benefit exchange between enterprises and local governments. In this way, we extend the theoretical boundary of home bias research and enrich the literature on home bias. Third, this paper has a certain practical significance. Our findings enable relevant suggestions and have implications for solving the current problems with labor relations and other related issues in China. Furthermore, other emerging markets can refer to this research to obtain helpful insights for guiding and promoting enterprises to fulfill employee social responsibility.

Reviewer’s Comment 5:

Please add study measures in the revised manuscript.

Authors’ Response (Comment 5):

We appreciate for the suggestion. As what the reviewer says, we did neglect the description of the research method, making the whole essay more difficult to understand. Therefore, we have added our research methods and measures of main variables in the part of introduction. Details are as follows: and is marked in blue in the revised manuscript (Line136-144 on p. 7) ：

We use the score index for employee social responsibility from the CNRDS dataset to measure corresponding corporate fulfillment, and we systematically investigate the effects of senior executives’ home bias on corporate employee social responsibility. The research methods detailed in this paper include theoretical analysis and empirical testing. We follow our empirical analysis with a series of robustness tests where propensity score matching (PSM) and the DID method are adopted. A cost‒benefit analysis of motivations is also carried out. Finally, the heterogeneous effects of senior executives’ home bias on employee social responsibility are analyzed via grouping regressions under different scenarios.

Reviewer’s Comment 6:

Conclusion: Please mention the limitations of the study and mention the future research direction as well.

Authors’ Response (Comment 6):

Thank you for the reviewer's comments. In the last part of conclusions, we have described limitations and future research directions of the study. However, due to our inaccurate descriptions, reviewer did not notice those illustrations. We are sorry for this misleading. Therefore, we have emphasized limitations and future research orientations in the revised manuscript in a more noticeable way. Specific minor adjustments are as follows, and is marked in blue in the revised manuscript (Line873-874 on p. 54) ：

The main limitations of this study and some future research directions are as follows: First, our work might be too narrow and insufficiently comprehensive to show that the origin place of senior executives is consistent enough with the firm-registered place to be used as the indicator for home bias. In future research, to improve measurement rationality and accuracy, a more comprehensive indicator should be built, probably by including the length of senior executives’ residency in their origin place, the religious culture thereof and other aspects. Second, there might be other potential channels through which senior executives’ home bias influences corporate behavior, in addition to the interest exchange view and the theory of identity we utilize in this paper. Future studies can thus explore these possible channels and further broaden the theoretical framework for research on the effects of senior executives’ home bias.

Reviewer’s Comment 7:

Proofread the manuscript to correct grammatical errors and smoothen the flow of contents.

Authors’ Response (Comment 7):

We greatly appreciate for the suggestions. Based on the reviewer's suggestion, we have revised the manuscript thoroughly and corrected grammatical errors to smoothen the flow of contents. Please review the revised manuscript.

---

## [Decision Letter · Decision Letter 1]

18 Nov 2022

Home Bias and Employee Social Responsibility: Identification vs. Benefit Exchange

PONE-D-22-15466R1

Dear Dr. Xing Rong,

We’re pleased to inform you that your manuscript has been judged scientifically suitable for publication and will be formally accepted for publication once it meets all outstanding technical requirements.

Kind regards,

Tai Ming Wut

Academic Editor

PLOS ONE

Additional Editor Comments (optional):

Reviewers' comments:

Reviewer's Responses to Questions

**Comments to the Author**

1. If the authors have adequately addressed your comments raised in a previous round of review and you feel that this manuscript is now acceptable for publication, you may indicate that here to bypass the “Comments to the Author” section, enter your conflict of interest statement in the “Confidential to Editor” section, and submit your "Accept" recommendation.

Reviewer #1: All comments have been addressed

Reviewer #2: All comments have been addressed

2. Is the manuscript technically sound, and do the data support the conclusions?

Reviewer #1: Yes

Reviewer #2: Yes

3. Has the statistical analysis been performed appropriately and rigorously? 

Reviewer #1: Yes

Reviewer #2: Yes

4. Have the authors made all data underlying the findings in their manuscript fully available?

Reviewer #1: Yes

Reviewer #2: Yes

5. Is the manuscript presented in an intelligible fashion and written in standard English?

Reviewer #1: Yes

Reviewer #2: Yes

6. Review Comments to the Author

Reviewer #1: (No Response)

Reviewer #2: Dear Authors,

My recommendations have been implemented. The work has a stable form and can be published

Best regards,

Reviewer

7. PLOS authors have the option to publish the peer review history of their article (what does this mean?). If published, this will include your full peer review and any attached files.

Reviewer #1: **Yes: **Dr. Shahid Ali

Reviewer #2: No

---

## [Editor Report · Acceptance letter]

30 Nov 2022

PONE-D-22-15466R1 

Home Bias and Employee Social Responsibility: Identification vs. Benefit Exchange 

Dear Dr. Rong:

I'm pleased to inform you that your manuscript has been deemed suitable for publication in PLOS ONE. Congratulations! Your manuscript is now with our production department. 

Kind regards, 

on behalf of

Dr. Tai Ming Wut 

Academic Editor

PLOS ONE